# Asymmetric framework motion of TCRαβ controls load-dependent peptide discrimination

Ana C Chang-Gonzalez[1], Robert J Mallis[2,3,4], Matthew J Lang[5,6], Ellis L Reinherz[3,4,7], Wonmuk Hwang[1,8,9]*

[1]Department of Biomedical Engineering, Texas A&M University, College Station, United States; [2]Department of Dermatology, Harvard Medical School, Boston, United States; [3]Laboratory of Immunobiology, Dana-Farber Cancer Institute, Boston, United States; [4]Department of Medicine, Oncology, Dana-Farber Cancer Institute, Boston, United States; [5]Department of Chemistry and Biomolecular Engineering, Vanderbilt University, Nashville, United States; [6]Department of Molecular Physiology and Biophysics, Vanderbilt University, Nashville, United States; [7]Department of Medicine, Harvard Medical School, Boston, United States; [8]Department of Materials Science & Engineering, Texas A&M University, College Station, United States; [9]Department of Physics & Astronomy, Texas A&M University, College Station, United States

*For correspondence:
hwm@tamu.edu

**Abstract** Mechanical force is critical for the interaction between an αβ T cell receptor (TCR) and a peptide-bound major histocompatibility complex (pMHC) molecule to initiate productive T-cell activation. However, the underlying mechanism remains unclear. We use all-atom molecular dynamics simulations to examine the A6 TCR bound to HLA-A*02:01 presenting agonist or antagonist peptides under different extensions to simulate the effects of applied load on the complex, elucidating their divergent biological responses. We found that TCR α and β chains move asymmetrically, which impacts the interface with pMHC, in particular the peptide-sensing CDR3 loops. For the wild-type agonist, the complex stabilizes in a load-dependent manner while antagonists destabilize it. Simulations of the Cβ FG-loop deletion, which reduces the catch bond response, and simulations with in silico mutant peptides further support the observed behaviors. The present results highlight the combined role of interdomain motion, fluctuating forces, and interfacial contacts in determining the mechanical response and fine peptide discrimination by a TCR, thereby resolving the conundrum of nearly identical crystal structures of TCRαβ-pMHC agonist and antagonist complexes.

## Editor's evaluation

Using extensive atomistic molecular dynamics simulations, the authors analyzed the TCR/pMHC interface with different peptide sequences and protein constructs. The results provide important insights into the catch-bond phenomenon in the context of T-cell activation. In particular, the analysis points to convincing evidence that supports the role of force in further discriminating different peptides during the activation process beyond structural considerations.

## Introduction

The αβ TCR (αβTCR) consists of the heterodimeric receptor TCRαβ formed by α and β chains each containing the pMHC-binding variable (V) and constant (C) domains (*Figure 1* and *Figure 2A*), and the noncovalently associated cluster of differentiation 3 (CD3) subunits that have cytoplasmic tails

**Figure 1.** Overview of figures and analyses in this work. *Figure 2* introduces the systems studied. WT systems were studied in three structural aspects (*Figures 3–5*). Similar analyses were done for the mutant systems (*Figures 6 and 7*). Appendix 1 and 2 provide simulations of additional mutants that we tested. We further examined the load- and time-dependent changes of the interfacial fit (*Figure 8*). All of the results collectively lead to the proposed mechanism of catch bond formation and ligand discrimination (*Figure 9*).

containing motifs for downstream signaling (*Rudolph et al., 2006*; *Wang and Reinherz, 2012*; *Brazin et al., 2018*). The TCR recognizes its cognate pMHC on the surface of an antigen-presenting cell (APC) at low or even single copy numbers from a pool of about $10^5$ different self-pMHC molecules (*Sykulev et al., 1996*; *Brameshuber et al., 2018*), while it also exhibits reactivity with certain closely related peptide variants, with similar or strikingly altered functional T-cell responses (*Ding et al., 1999*; *Hausmann et al., 1999*; *Lee et al., 2004*; *Borbulevych et al., 2009*; *Baker et al., 2012*; *Birnbaum et al., 2014*). Considering the μM to hundreds of μM TCRαβ-pMHC equilibrium binding affinity (*Wang and Reinherz, 2012*), several models have been proposed to account for the exquisite specificity and sensitivity of the αβTCR (*Chakraborty and Weiss, 2014*; *Brazin et al., 2015*; *Schamel et al., 2019*; *Zhu et al., 2019*; *Mariuzza et al., 2020*; *Liu et al., 2021*).

A critical factor for peptide discrimination is physiological force applied to the TCRαβ-pMHC complex (*Reinherz et al., 2023*). A cognate peptide antigen elicits a catch bond behavior where the TCRαβ-pMHC bond lifetime increases with force that peaks in the 10–20 pN range, and is observed with the clonotypic ligand-binding TCRαβ heterodimer in isolation or with the holoreceptor αβTCR including the non-covalently associated CD3 signaling subunit dimers (CD3ϵγ, CD3ϵδ, and CD3 ζ ζ). The catch bond is coupled with a roughly 10 nm structural transition in both (*Das et al., 2015*; *Das et al., 2016*; *Banik et al., 2021*), which supports the notion that the αβTCR acts as a mechano-sensor (*Kim et al., 2009*; *Kim et al., 2012*; *Brazin et al., 2015*; *Brazin et al., 2018*; *Choi et al., 2023*; *Reinherz et al., 2023*). In our previous molecular dynamics (MD) study (*Hwang et al., 2020*), instead of enforcing dissociation of the complex with high force, as done in steered MD simulations (*Sibener et al., 2018*; *Wu et al., 2019*), we applied pN-level forces and examined the behavior of the JM22 TCR complexed with an HLA-A*02:01 molecule presenting a peptide from an influenza virus matrix protein. We found that the TCRαβ-pMHC complex is in a loosely-bound state in the absence of load, which allows domain motion. Application of a 16-pN force suppresses the motion and overall enhances the fit among domains. We proposed a model where the TCRαβ-pMHC catch bond arises due to stabilization of the interface by altering the conformational motion of TCRαβ.

An important question regards the generality of this dynamic mechanism in other TCRs. To this end, we study the A6 TCR, which recognizes the Tax peptide (LLFGYPVYV) of the human T lympho-tropic virus type 1 (*Garboczi et al., 1996b*) bound to HLA-A*02:01, the same MHC as for JM22. We perform all-atom MD simulations with the Tax peptide (wild type; WT) (*Garboczi et al., 1996a*) and four mutant peptides with a single-residue substitution: Y5F (*Scott et al., 2011*), V7R, P6A, and Y8A (*Ding et al., 1999*). Below, we call the TCRαβ-pMHC complex by the name of the corresponding peptide. For example, Y5F refers to the complex with the Y5F peptide (PDB 3QFJ, *Figure 2C*).

While the crystallographic structures of these complexes are very similar (*Ding et al., 1999*; *Scott et al., 2011*; *Figure 2C*), they differ in immunogenicity. P6A and Y8A are weak antagonists because they inhibit T-cell function only at 1000-times higher molar concentration than that needed by the WT

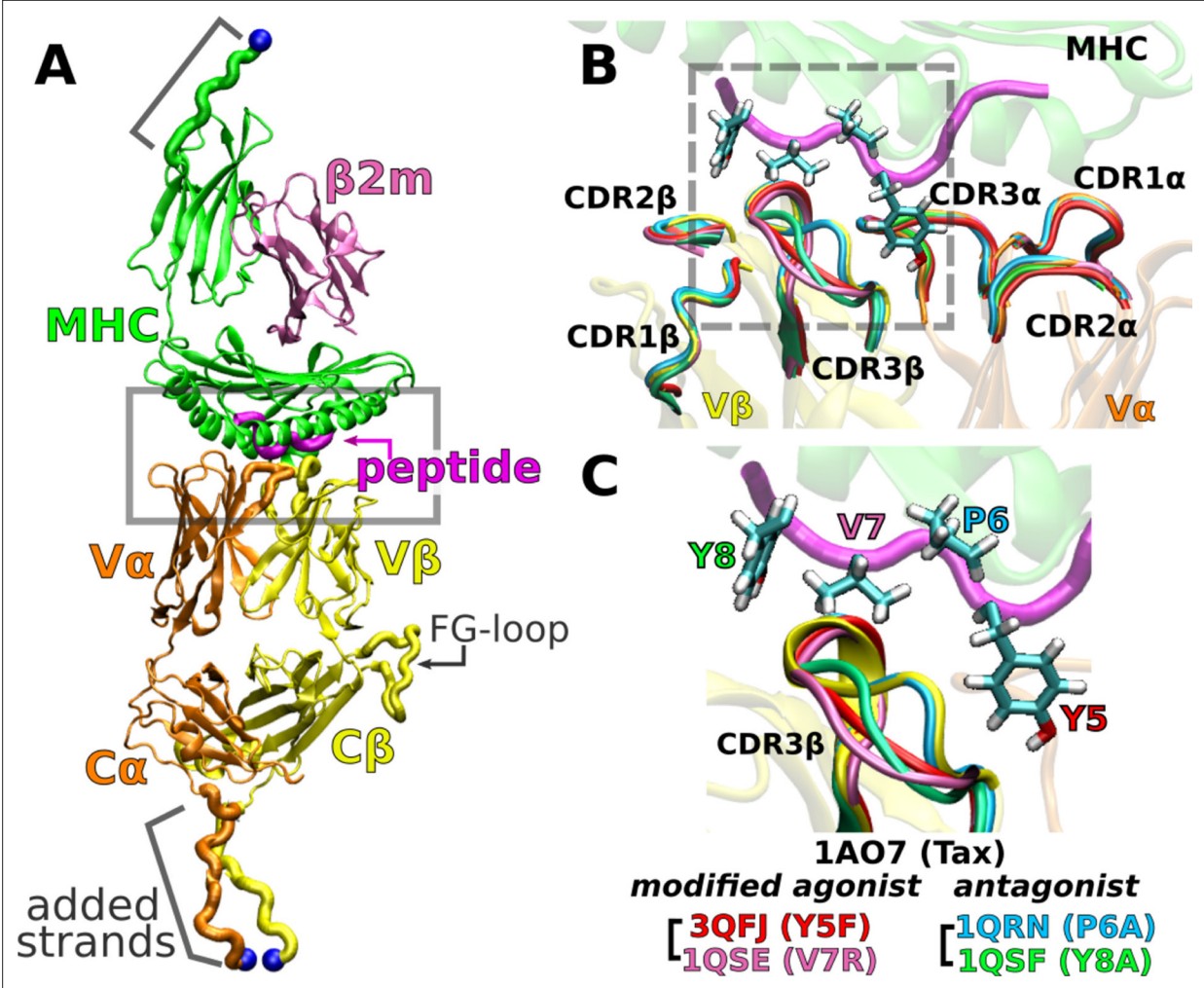

**Figure 2.** A6 TCRαβ-pMHC complex. (**A**) WT (Protein Data Bank, PDB 1AO7). The missing Cα domain was added based on PDB 1QSE (Structure preparation). Blue spheres: terminal Cα atoms held at set extensions during the simulation (*Table 1*). β2m: β2 microglobulin. (**B**) Overlay of the X-ray structures of the WT and four point mutants of the Tax peptide at the boxed region of panel A. The CDR loops take nearly identical conformations in different structures. (**C**) Magnified view of the dashed box in panel B, focusing on the conformation of CDR3β. PDB names for A6 TCRαβ-pMHC complexes containing mutant peptides are listed.

for activation (*Hausmann et al., 1999*; *Ding et al., 1999*). We refer to them simply as 'antagonists.' Y5F is similar to WT in terms of equilibrium binding affinity and T-cell activation in vitro (*Hausmann et al., 1999*; *Scott et al., 2011*). V7R induces effector functions comparable to WT at 10- to 100-times higher concentrations (*Ding et al., 1999*). We call Y5F and V7R as 'modified agonists.' There have been several experimental and computational studies comparing the effects of peptide modifications or pMHC binding on A6 TCR (*Baker et al., 2000*; *Michielin and Karplus, 2002*; *Davis-Harrison et al., 2005*; *Cuendet and Michielin, 2008*; *Borbulevych et al., 2009*; *Cuendet et al., 2011*; *Scott et al., 2011*; *Ayres et al., 2016*; *Fodor et al., 2018*), but load was not explicitly considered. To simulate a complex under load, we held the distance between the terminal Cα atoms of the complex (*Figure 2A*) at a set extension for the duration of the simulation. This was done by applying harmonic positional restraints so that the terminal Cα atoms were allowed to fluctuate in position, hence resulting in instantaneous fluctuation in the applied force akin to loading in experiments. We refer to a simulation as either low or high load based on the average load, which was around the physiological 10–20 pN range (*Table 1*). To our knowledge, the present study is the first to elucidate the dynamic mechanism of the A6 complex harboring different peptides under load.

We found that differences between the WT and peptide mutants lie in dynamic responses to applied load. In the WT, physiological level load stabilized the TCRαβ-pMHC interface as well as the

**Table 1.** Simulations of TCRαβ-pMHC complexes.

Load is average after 500 ns (See Selecting extensions in Computational methods). Parentheses after the average load show standard deviation (std) in forces measured in 40-ns intervals after 500 ns. The std is approximately proportional to the average force, which is a consequence of the positional restraints for applying load being in thermal equilibrium at constant temperature. Exceptions were dFG$^{low}$ and Y8A$^{low}$. They had larger std relative to the average force due to the extra motion caused by the unstable TCRαβ-pMHC interface (see Appendix 3).

| Peptide | PDB | Extension (Å) | Time (ns) | Load (pN) | Label | Description |
|---|---|---|---|---|---|---|
| Tax (WT) | 1AO7 | – | 1170 | – | WT$^0$ | |
| | | 182.6 | 1350 | 13.2 (5.65) | WT$^{low}$ | wild-type |
| | | 187.7 | 1250 | 18.2 (9.17) | WT$^{high}$ | |
| Tax (dFG-pMHC) | 1AO7 | 180.5 | 1100 | 14.9 (11.2) | dFG$^{low}$ | dFG (*Table 2*) with pMHC |
| | | 188.9 | 1100 | 29.0 (11.8) | dFG$^{high}$ | |
| Y5F | 3QFJ | – | 1180 | – | Y5F$^0$ | |
| | | 181.4 | 1200 | 8.24 (2.46) | Y5F$^{low}$ | |
| | | 186.2 | 1200 | 23.7 (9.16) | Y5F$^{high}$ | |
| V7R | 1QSE | – | 1020 | – | V7R$^0$ | modified agonists |
| | | 177.5 | 1012 | 10.3 (3.79) | V7R$^{low}$ | |
| | | 186.2 | 1003 | 17.8 (7.55) | V7R$^{high}$ | |
| P6A | 1QRN | – | 1090 | – | P6A$^0$ | |
| | | 175.2 | 1018 | 8.81 (2.77) | P6A$^{low}$ | |
| | | 186.0 | 1020 | 13.5 (6.20) | P6A$^{high}$ | weak antagonists |
| Y8A | 1QSF | – | 1020 | – | Y8A$^0$ | |
| | | 176.5 | 1280 | 12.0 (9.79) | Y8A$^{low}$ | |
| | | 187.4 | 1330 | 18.1 (6.79) | Y8A$^{high}$ | |

subdomain motion within TCRαβ. Modified agonists maintained stable contacts, yet high loads led to destabilization. Antagonists had less stable interfaces under load as the mutated residues disrupted surrounding interfacial contacts. Motion within the TCR, such as the Vα-Vβ scissoring as observed in *Hwang et al., 2020*, and an asymmetric bending of the V-module relative to the C-module, were coupled to the interactions between the variable domains and pMHC such that a single-residue mutation in the peptide affected the conformational behavior of the whole TCR. The present results suggest that the conserved TCRαβ framework motion is leveraged when determining the mechanically matched pMHC, a mechanism that is broadly applicable to different TCRαβ systems.

## Results

We first study the WT-based systems to gain insight into the functional implications of TCRαβ-pMHC structural dynamics, followed by point mutations in the Tax peptide. Our analyses below involve time-dependent inter-domain contact dynamics, domain motion, and their dependence on applied load. *Figure 1* provides an overview of other figures and analyses, as a guide for navigating this study.

### Applying loads to TCRαβ-pMHC complexes

To apply load, we used harmonic positional restraints on the terminal C$_α$ atoms at given extensions (*Figure 2A*; see Laddered extension with added strands in Computational methods). This was based on the realistic situation of immune surveillance. When a T-cell interacts with an APC, other molecules such as CD2 and CD58 maintain the separation near the ~120 Å span of the TCRαβ-pMHC complex (*Reinherz et al., 2023*). The force applied to the complex fluctuates via thermal motion and through cellular activities such as coupling to the actomyosin machinery within the T-cell and APC (*Liu*

et al., 2016; Feng et al., 2017; Reinherz et al., 2023). Restraining terminal $C_\alpha$ atoms in simulation mimics the membrane anchoring of these molecules with a relatively constant spacing and fluctuating force. The measured instantaneous force varies in magnitude and direction across coordinate frames. Averaged over time, it becomes mainly longitudinal (Table 1). Despite the temporal variation in the applied force, the results below show that the response of the complex depends on the average force and the ligand.

In simulation, soft harmonic positional restraint can be used for conformational sampling (Pitera and Chodera, 2012). Rather than positional sampling, our goal is applying force at a given extension, for which we used a stiff 1-kcal/[mol.Å$^2$] potential (Laddered extension with added strands) that yields thermal fluctuations of amplitude ~0.8 Å (Hwang, 2007). While it is also possible to apply a constant force (Gomez et al., 2021; Stirnemann, 2022), it would not reflect the membrane-anchored state. In addition, the extension of the complex will fluctuate under a constant force, and the actual load that the TCRαβ-pMHC interface experiences will likewise fluctuate. How load propagates through the complex and distributes across the interface is a subject of a future study.

## Load stabilizes WT TCRαβ-pMHC interfacial contacts

We assessed the effect of load on WT first by counting high-occupancy contacts with pMHC (Figure 3A; see Contact analysis in Computational methods). WT$^{low}$ had the least number of contacts, followed by WT$^0$ and WT$^{high}$, indicating low and high loads may have opposite effect on the interfacial stability. Vαβ-pMHC without the C-module formed the most contacts. This indicates that without a proper load, the C-module is detrimental to the stability of the interface with pMHC, as we found previously for the JM22 TCR (Hwang et al., 2020).

Time-dependent changes in the interfacial contacts were monitored using the Hamming distance $\mathcal{H}$ (Hamming, 1950; Hwang et al., 2020). $\mathcal{H}$ is the number of the initial high-occupancy contacts (those with greater than 80% average occupancy during the first 50 ns) that are subsequently lost during the simulation. A low $\mathcal{H}$ means that such contacts persist while a high $\mathcal{H}$ means the corresponding number of initial high-occupancy contacts are missing. Consistent with the contact count, $\mathcal{H}$ remained low for WT$^{high}$ and Vαβ-pMHC (Figure 3B). In WT$^0$ and WT$^{low}$, $\mathcal{H}$ increased after about 500 ns and 900 ns, respectively. Thus, the relatively high number of interfacial contacts for WT$^0$ (Figure 3A) is due to the formation of new contacts rather than by maintaining the initial contacts.

Occupancy heat maps provide the time dependence of individual contacts. For WT$^{high}$ and Vαβ-pMHC, high-occupancy contacts persist throughout the simulation (blue regions in Figure 3C and Figure 3—figure supplement 1) while WT$^0$ or WT$^{low}$ exhibited breakage of contacts, especially when $\mathcal{H}$ increased (dashed arrows in Figure 3D and E). Differences in the interfacial contacts also manifest in their location. We displayed the backbone $C_\alpha$ atoms of Vα and Vβ residues that form contacts with pMHC with greater than 80% average occupancy (averaging was done after the initial 500 ns; Figure 3F). In WT$^0$, contacts are spread apart, and in WT$^{low}$ they lie mostly along the length of the peptide. These layouts potentially make interfacial contacts more prone to break via easier access by water molecules. In WT$^{high}$ and Vαβ-pMHC, high-occupancy contacts form more compact clusters. Exposure to water of the TCRαβ residues involved in those contacts was measured by their buried surface area (BSA), which follows the same trend as the number of high occupancy contacts (Figure 3A vs. Figure 3G) Furthermore, this trend also applied to the root-mean square fluctuation (RMSF) of Tax peptide backbone $C_\alpha$ atoms, even though RMSF values were small (Figure 3H).

Experimentally, the WT complex has a relatively strong affinity as a TCRαβ (about 1 μM) (Ding et al., 1999), which may be why the interface with pMHC involved more contacts in WT$^0$ compared to WT$^{low}$. At low load, the short distance between restraints on the ends of the complex (Figure 2A) allows wider transverse motion that in turn generates a shear stress or a bending moment at the interface. Transverse stress will be less for WT$^0$ where the end moves freely, and for WT$^{high}$ where lateral motion is suppressed. The extent of transverse motion can be seen by the RMSF of the center of mass of the $C_\alpha$ atoms of the peptide in the transverse direction, which was 16.3 Å for WT$^{low}$ and 12.7 Å for WT$^{high}$. The high stability of Vαβ-pMHC and WT$^{high}$ agree well with the results for JM22 (Hwang et al., 2020).

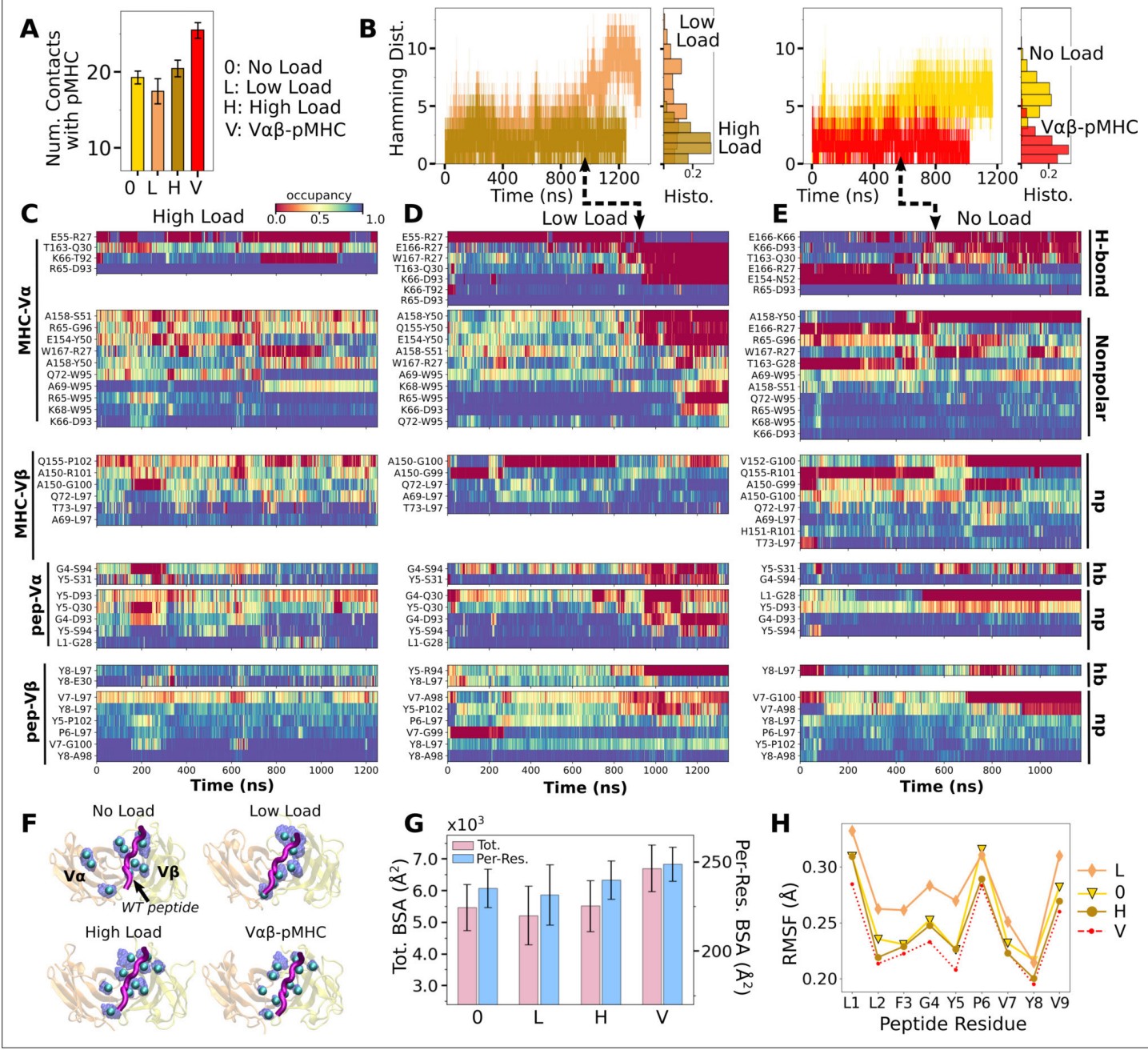

**Figure 3.** Load dependence of the WT TCRαβ-pMHC interface. (**A**) Number of high-occupancy contacts (Contact analysis). Bars: std. (**B**) Hamming distance $\mathcal{H}$ over time. Histograms are for the interval after 500 ns. Dashed arrows mark increase in $\mathcal{H}$, corresponding to contacts lost. (**C–E**) Contact occupancy heat maps for the interface with pMHC. H-bond/hb: hydrogen bonds, including salt bridges, and np: nonpolar (Contact analysis). (**F**) Location of $C_\alpha$ atoms of the residues whose contacts with pMHC have greater than 80% average occupancy. Cyan spheres: last frame of simulation. Transparent blue: locations rendered every 0.2 ns showing positional variability. (**G**) Total (pink) and per-residue (blue) BSA for interfacial residues with greater than 80% maximum instantaneous occupancy (BSA). Bars: std. (**H**) RMSF of backbone $C_\alpha$ atoms of the peptide after 500 ns. For calculation, the $C_\alpha$ atoms were aligned to those at the beginning of the production run.

The online version of this article includes the following figure supplement(s) for figure 3:

**Figure supplement 1.** Contact occupancy heat maps for Vαβ-pMHC.

**Table 2.** Simulations of truncated structures from PDB 1AO7.

| Label | Modification | Time (ns) |
|---|---|---|
| V$\alpha\beta$ | V$\alpha$-V$\beta$ only (no pMHC) | 1060 |
| T$\alpha\beta$ | TCR$\alpha\beta$ only (no pMHC) | 1000 |
| V$\alpha\beta$-pMHC | V$\alpha\beta$ with pMHC (no C-module) | 1020 |
| dFG | T$\alpha\beta$ without the C$\beta$ FG-loop (no pMHC) | 1000 |

## Influence of pMHC and load on Vα-Vβ motion

We analyzed the motion between Vα and Vβ (Vα-Vβ motion) to find its effect on the TCRαβ-pMHC interface. Compared to the unliganded systems (Vαβ and Tαβ; *Table 2*), the number of high-occupancy Vα-Vβ contacts increased slightly in Vαβ-pMHC ('V' in *Figure 4A*), while it decreased in full TCRαβ-pMHC complexes ('0', 'Low', and 'High' in *Figure 4A*). This shows that the Vα-Vβ interface is difficult to organize with the restrictions imposed by the bound pMHC, except in the absence of the constant domains. The number of Vα-Vβ contacts in the liganded systems is the smallest for WT[low], similar to the case for the number of contacts with pMHC (*Figure 3A*), again reflecting a destabilizing effect with low load.

To measure the Vα-Vβ motion, coordinate trajectories were first aligned to the stably folded β-sheet cores of the two domains. Triads were then assigned to the cores and principal component analysis (PCA) was performed on the triad trajectories (*Figure 4B*; Variable domain triads and PCA; *Hwang et al., 2020*). Since triads were assigned in the same way, the relative Vα-Vβ motion in different simulations can be studied by comparing their PCA.

The amplitude of PC1 is lower when the number of Vα-Vβ contacts is higher (*Figure 4A* vs. *Figure 4—figure supplement 1A*). Directions of PCs differed to varying extents (arrows in *Figure 4—figure supplement 1B*). Similarity of the directions was measured by the absolute value of the dot product between PCs as 18-dimensional unit vectors (for the six arms from two triads) in different systems. A value of 1 corresponds to the same direction, and 0 means an orthogonal direction (*Figure 4—figure supplement 1C*). For PC1, a high degree of similarity was observed between Tαβ and Vαβ, which is consistent with their similarity in the number of Vα-Vβ contacts (*Figure 4A*) and PC amplitudes (*Figure 4—figure supplement 1A*). Among triad systems with bound pMHC, WT[low] differed significantly in the PC1 direction compared to others (*Figure 4—figure supplement 1C*, darker colors). The dot products varied more for PC2 and PC3, which capture finer motions with smaller amplitudes (*Figure 4—figure supplement 1C*).

To determine how the Vα-Vβ motion influences the interface with pMHC, we measured the distance between CDR3 loops (CDR3 distance), which play a central role in peptide discrimination (*Figure 2B and C* and *Figure 4—figure supplement 2A–C*). Unliganded Tαβ and Vαβ had greater fluctuation in the CDR3 distance (larger std), as they are unrestrained by pMHC. Among the pMHC-bound systems, WT[high] and Vαβ-pMHC had small CDR3 distance (averages of 10.3 and 10.5 Å, respectively; *Figure 4—figure supplement 2B C*). CDR3 distance was larger for WT[0] (11.3 Å), which reflects an altered interface with pMHC. For WT[low], the CDR3 distance varied more widely, with more than a 2-fold increase in standard deviation. The increase in CDR3 distance of WT[low] happens after the increase in $\mathcal{H}$ (800 ns; *Figure 3B and D* and *Figure 4—figure supplement 2C*), suggesting a loss of contacts at the interface is related to the Vα-Vβ motion.

PCA decomposes the Vα-Vβ motion into mutually orthogonal directions. We made 2-dimensional histograms of each of these projections versus the corresponding CDR3 distance (*Figure 4—figure supplement 1D*). If any of the PC modes is strongly correlated with the CDR3 distance, the corresponding histogram would exhibit a slanted profile. However, no clear correlation could be seen (*Figure 4—figure supplement 1D*), suggesting that the changes in CDR3 distance may depend on combinations of PCs. We addressed this possibility by considering angles between matching arms of the two triads that do not rely on PCA (*Figure 4B*). The $\mathbf{e}_1$-$\mathbf{e}_1$ angle, herein called $\angle e_1$ (and similarly define $\angle e_2$ and $\angle e_3$; *Figure 4B*), can change either by the $\mathbf{e}_1$ arms turning up and down ('flap') or in and out of the page ('twist') in *Figure 4B*. Angles $\angle e_2$ and $\angle e_3$ depend primarily on rotation indicated by dashed arrows in *Figure 4B* ('scissor') (*Hwang et al., 2020*).

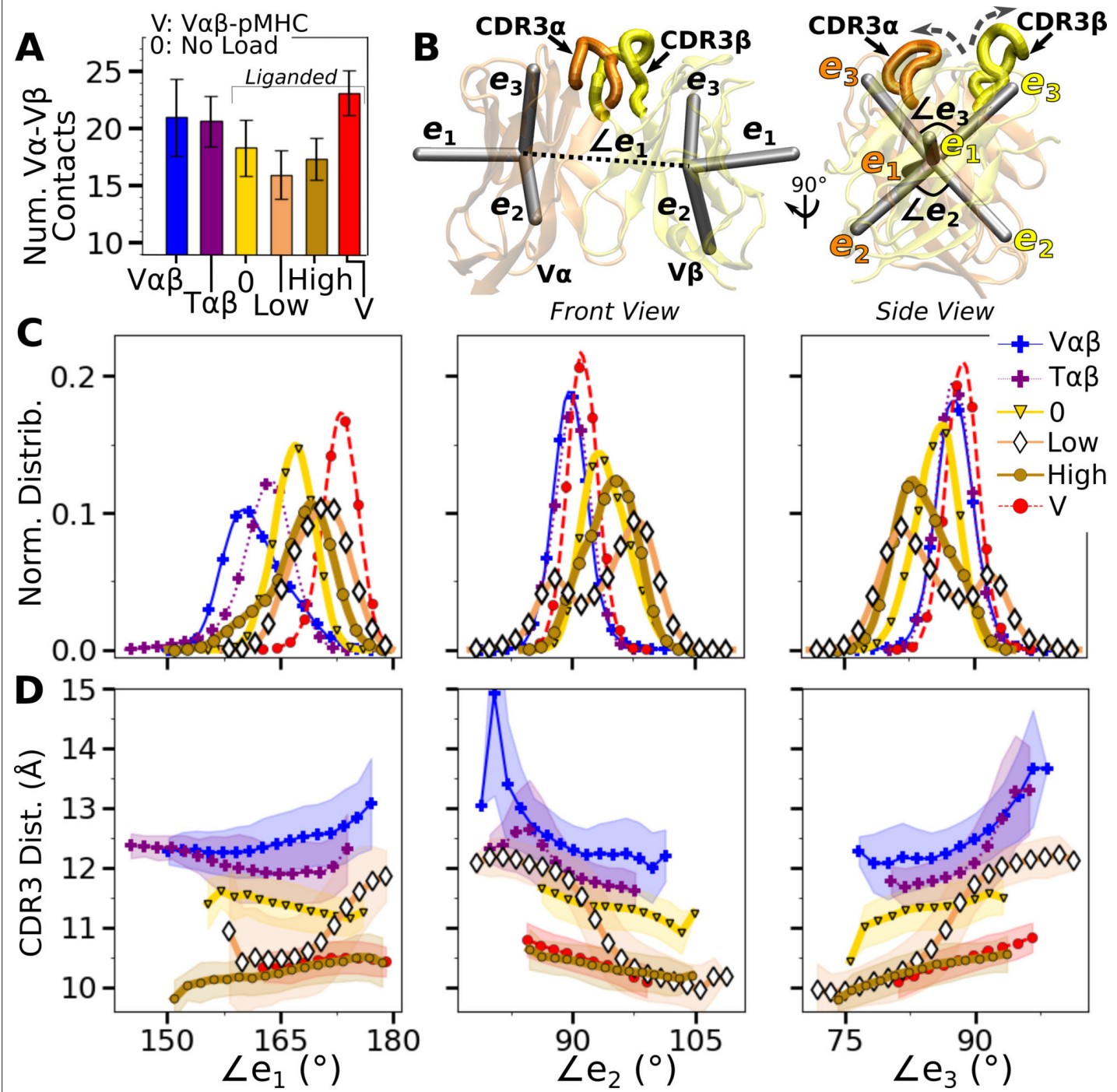

**Figure 4.** Vα-Vβ motion. (**A**) Number of high-occupancy contacts (Contact analysis). Bars: std. (**B**) Triads {$\mathbf{e}_1, \mathbf{e}_2, \mathbf{e}_3$} assigned to each domain. Angles between triad arms ($\angle e_1$, $\angle e_2$, and $\angle e_3$) are marked. CDR3 loops are represented as thick tubes. Dashed arrows indicate directions where the CDR3 distance increases via the scissor motion. (**C**) Histograms of the 3 angles between the triad arms. For WT[low], the smaller peaks in distributions of $\angle e_2$ and $\angle e_3$ arise from simulation trajectories after 1 µs. (**D**) CDR3 distance vs. the 3 angles. Transparent band: std of the CDR3 distance in each bin. Statistics for bins deteriorate in large- or small-angle tails that contain very few frames.

The online version of this article includes the following figure supplement(s) for figure 4:

**Figure supplement 1.** PCA of Vα-Vβ motion.

**Figure supplement 2.** Trajectories of the V-module motion.

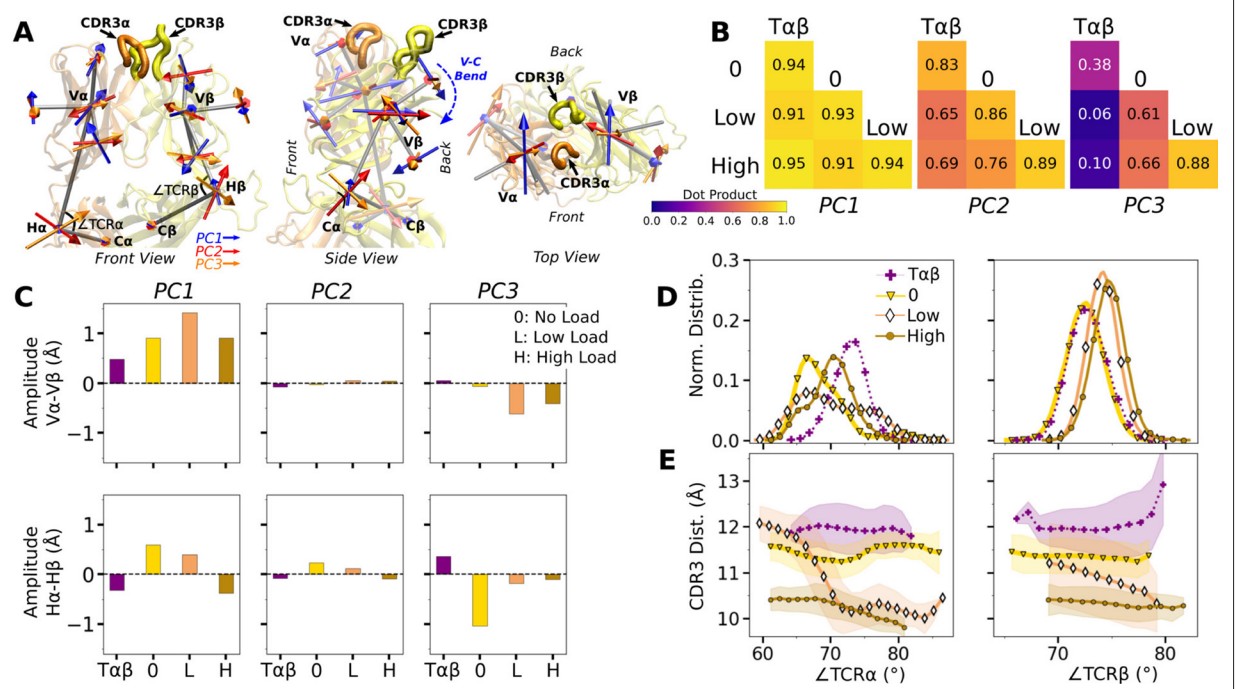

**Figure 5.** WT V-C dynamics. (**A**) Average BOC built from the unliganded Tαβ. The C-module is used as orientational reference, hence PC arrows are visible only for the V-module and hinges. (**B**) Dot products computed between the BOC PCs in listed systems. Values closer to 1.0 denote similar V-C BOC direction of motion. (**C**) Difference in amplitude between the α and β chain motion measured between Vα and Vβ (top), and Hα and Hβ (bottom). PC amplitudes are proportional to the lengths of the arrows in panel A. (**D**) Histograms of hinge angles (defined in panel A) for each chain. (**E**) CDR3 distance vs. hinge angles.

The online version of this article includes the following figure supplement(s) for figure 5:

**Figure supplement 1.** V-C PC amplitude and contacts.

Histograms of the three angles (*Figure 4C*) show a clearer difference than individual PCs among the systems tested, and the CDR3 distance varies with the angles (*Figure 4D*). The wider distributions for angles of WT[low] and WT[high] reflect their higher PC amplitudes (*Figure 4—figure supplement 1A*). The symmetric distributions of $\angle e_2$ and $\angle e_3$ can be seen from the two peaks for WT[low], which is due to the reciprocal behavior of the scissoring motion involving the two angles (*Figure 4B*, side view, and *Figure 4C*, open diamonds). The two peaks are also related to the changes in the CDR3 distance (*Figure 4D*), which reflects an agitating effect of the mild load on the scissor motion. Given the definitions of the angles, the CDR3 distance will increase (dashed arrows in *Figure 4B*) with larger $\angle e_1$ or $\angle e_3$, or with smaller $\angle e_2$ (*Figure 4D*). WT[0], despite the apparent stability of the interface, had a larger CDR3 distance than WT[high] and Vαβ-pMHC, again indicating a disrupted interface. These results show that the CDR3 motion is coupled to the Vα-Vβ motion, especially the scissoring motion.

## Asymmetric V-C motion influences the load response of the complex

We next analyzed the motion of the V-module relative to the C-module (V-C motion). The number of high-occupancy Cα-Cβ contacts did not vary significantly (in the 31–34 range) and they were more than the number of Vα-Vβ contacts, similar to the case for the JM22 TCR (*Hwang et al., 2020*). The C-module thereby influences the V-module as a single unit. The V-C motion was analyzed by performing PCA on the bead-on-chain (BOC) model constructed based on the β-sheet core of each domain, and hinges between V- and C-domains denoted as Hα and Hβ (*Figure 5A*; V-C BOC and PCA). Coordinate trajectories were aligned to the C-module so that the motion of the V-module relative to the C-module in different simulations can be compared. Across different systems, amplitudes of PCs were similar (*Figure 5—figure supplement 1*). PC1 (V-C bend; *Figure 5A*) was similar among systems, as seen by the values of dot products being close to 1.0 (*Figure 5B*). Directions of higher PCs varied more, similarly as higher PCs for the Vα-Vβ motion.

We noticed that Vα bends more compared to Vβ, as can be gleaned from the longer PC arrows for Vα (*Figure 5A*). We quantified this asymmetry by subtracting the amplitudes of motion for domains in the β chain from those for the matching domains in the α chains, where positive or negative values respectively indicate greater or less motion of the α compared to the β chain (*Figure 5C*). Compared to Tαβ, binding of pMHC increases the α-chain motion, which is the greatest in WT$^{low}$ (*Figure 5C*, PC1 in top row). The greater degree of Vα-Cα motion is consistent with the smaller number of Vα-Cα contacts compared to Vβ-Cβ (*Figure 5—figure supplement 1*).

The asymmetry was further analyzed by measuring hinge angles ∠TCRα and ∠TCRβ independently of PCA (*Figure 5A*). Distributions of ∠TCRα varied more compared to ∠TCRβ (*Figure 5D*). A wide distribution of ∠TCRα for WT$^{low}$ is related to the increase in the CDR3 distance and concomitant changes in the triad arm angles later during the simulation (*Figure 4—figure supplement 2C*). For WT$^{low}$ and WT$^{high}$, CDR3 distance decreases with increasing hinge angles, especially with ∠TCRα (*Figure 5E*), which suggests that unbending of the V-module under load helps with bringing the CDR3 loops closer together. In WT$^{low}$, this state is not maintained and ∠TCRα decreases (more bending) as the CDR3 distance increases (*Figure 5E*) which happens after the increase in $\mathcal{H}$ (*Figure 3B*). These results suggest a mechanism by which the asymmetric response of the whole TCRαβ to load affects the binding with pMHC by controlling the relative positioning between the CDR3 loops via the Vα-Vβ motion. For this, the Cβ FG-loop plays a critical role, as simulations of the bound complex without the Cβ FG-loop resulted in a smaller ∠TCRβ and an over-extended ∠TCRα (see Appendix 1).

## Effects of point mutations on the peptide

In the WT crystal structure, the side chain of Y5 in the Tax peptide is located between the CDR3 loops of Vα and Vβ while V7 mainly contacts CDR3β (*Figure 2C*). P6 makes one contact with CDR3β. The side chain of Y8 is located between CDR3β and the α1 helix of MHC. Crystal structures of point mutants of these four residues are very similar in terms of interfacial contacts, docking angle, and CDR loop conformations, with the only structurally observable difference located at CDR3β (*Figure 2B and C*; *Ding et al., 1999*; *Scott et al., 2011*). However, point mutations profoundly affect dynamics of the complex, as explained below.

Modified agonists Y5F and V7R had about the same number of contacts with pMHC as the WT complexes, but high load resulted in fewer contacts, indicating a potential slip bond behavior (*Figure 6A*), although loss of contacts in Y5F$^{high}$ might have been due to a higher load experienced compared to other complexes at the same extension (23.7 pN; *Table 1*). Antagonists P6A and Y8A had overall fewer contacts with pMHC without a consistent load dependence (*Figure 6A*). This trend was also seen in BSA profiles of residues forming high-occupancy contacts with pMHC (*Figure 6B*). For modified agonists, higher load also resulted in greater increase of $\mathcal{H}$, whereas the trend was opposite for antagonists (*Figure 6—figure supplement 1*). The large number of contacts with pMHC for modified agonists (*Figure 6A*) despite an increase in $\mathcal{H}$ suggests an altered binding rather than maintaining the initial contacts.

In contact heat maps, the Y5 residue of the WT peptide forms a hydrogen bond with αS31 and nonpolar contacts with a few residues in both Vα and Vβ (*Figure 3C–E*, *Figure 6C*). In Y5F, the hydrogen bond with αS31 cannot form, and many of the nonpolar contacts with F5 break under load later during the simulation (*Figure 6—figure supplement 2A*). The breakage coincides with the increase in $\mathcal{H}$ (*Figure 6—figure supplement 1A*). In addition, contacts involving Y8 and V7 also break in Y5F$^{high}$ (*Figure 6—figure supplement 2A*). Thus, the Y5-αS31 hydrogen bond may stabilize the interface with pMHC by arranging other nearby residues to form nonpolar contacts in favorable positions; its absence would make the nonpolar contacts more prone to break under load. The relative stability of Y5F$^{0}$ can also be seen by the similarity in the locations of high-occupancy contact residues between WT and Y5F$^{0}$ (*Figure 3F* vs. *Figure 6—figure supplement 3A*). Experimentally, Y5F has kinetic and cytotoxicity profiles similar to WT (*Hausmann et al., 1999*; *Scott et al., 2011*). Its dependence on load needs further experimental analysis. On the other hand, V7 of the WT peptide forms nonpolar contacts with residues in CDR3β (*Figure 3C–E*, *Figure 6C*). In V7R, nonpolar contacts with CDR3β form with reduced occupancy, and contacts involving Y8 are also disrupted (*Figure 6C* vs. *Figure 6—figure supplement 2B*).

For antagonists, more contacts broke, which again involve non-mutated residues such as Y5 and V7 (*Figure 6D–F*, *Figure 6—figure supplement 2C*). The greater number of contacts in Y8A$^{high}$ compared

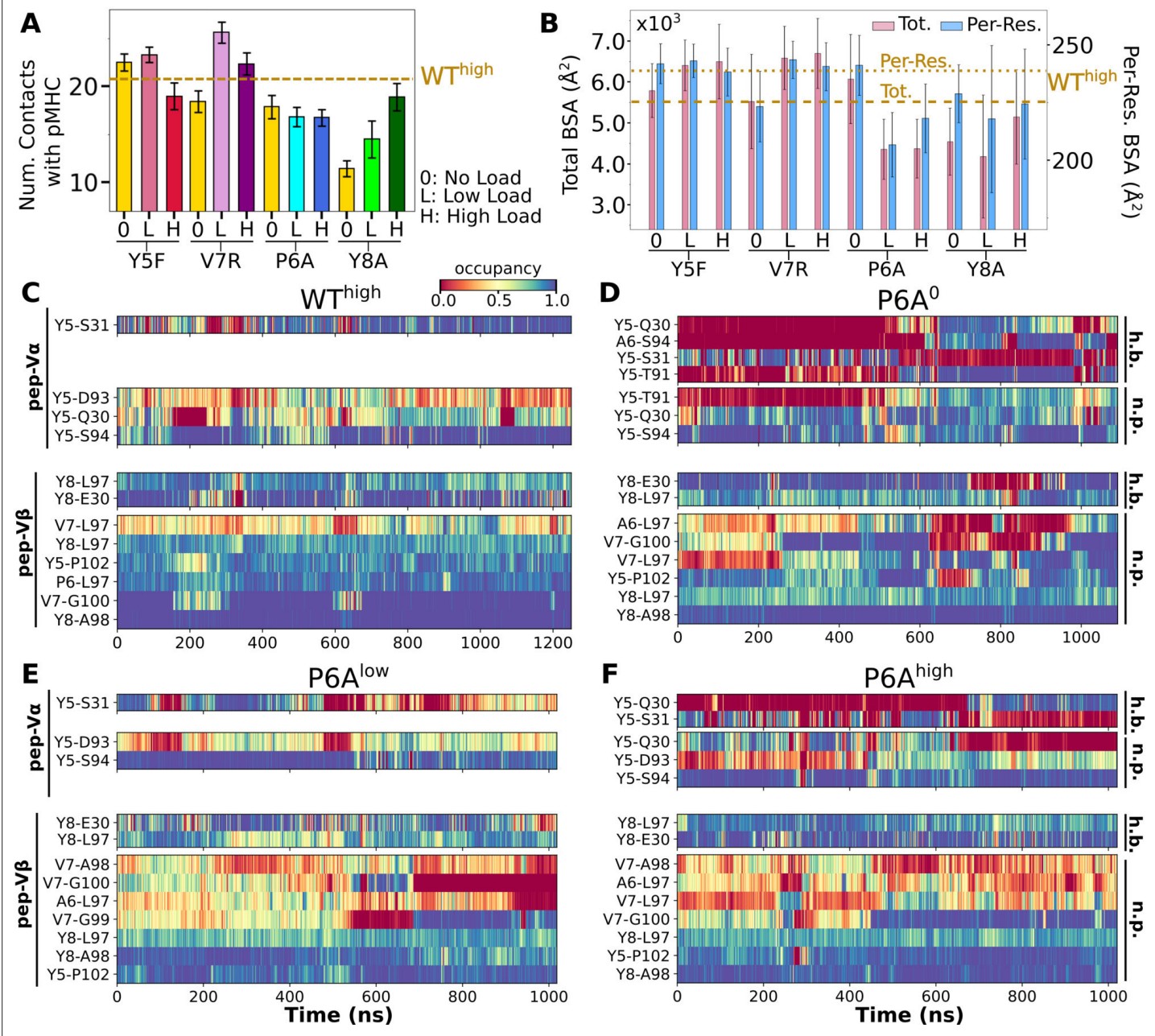

**Figure 6.** Interface with pMHC containing mutant peptides. The same occupancy cutoffs were used as in *Figure 3*. (**A**) Number of contacts with pMHC. Dashed line: count for WT[high] in *Figure 3A*, for reference. (**B**) Total (pink) and per-residue (blue) BSA. Dashed and dotted lines: values for WT[high] (*Figure 3G*). (**C–F**) Contact heat maps for peptide residues 5–8. (**C**) WT[high] (included in *Figure 3C*), and (**D**) P6A[0], (**E**) P6A[low], and (**F**) P6A[high].

The online version of this article includes the following figure supplement(s) for figure 6:

**Figure supplement 1.** Trajectories of $\mathcal{H}$ for mutant complexes.

**Figure supplement 2.** Contact occupancy heat maps for residues 5–8 of the mutant peptides.

**Figure supplement 3.** Locations of high-occupancy contacts with pMHC in mutant systems.

to Y8A[0] and Y8A[low] (*Figure 6A*) despite smaller number of contacts involving key peptide residues Y5–A8 (*Figure 6—figure supplement 2C*) suggests formation of additional contacts with other parts of MHC as a result of an altered interface. Experimentally, binding of the A6 TCR to pMHC containing the P6A or Y8A peptide was not detected in vitro (*Ding et al., 1999*). Thus, Y8A in principle could exhibit a catch bond, but forming the complex in the loaded state may be kinetically inaccessible.

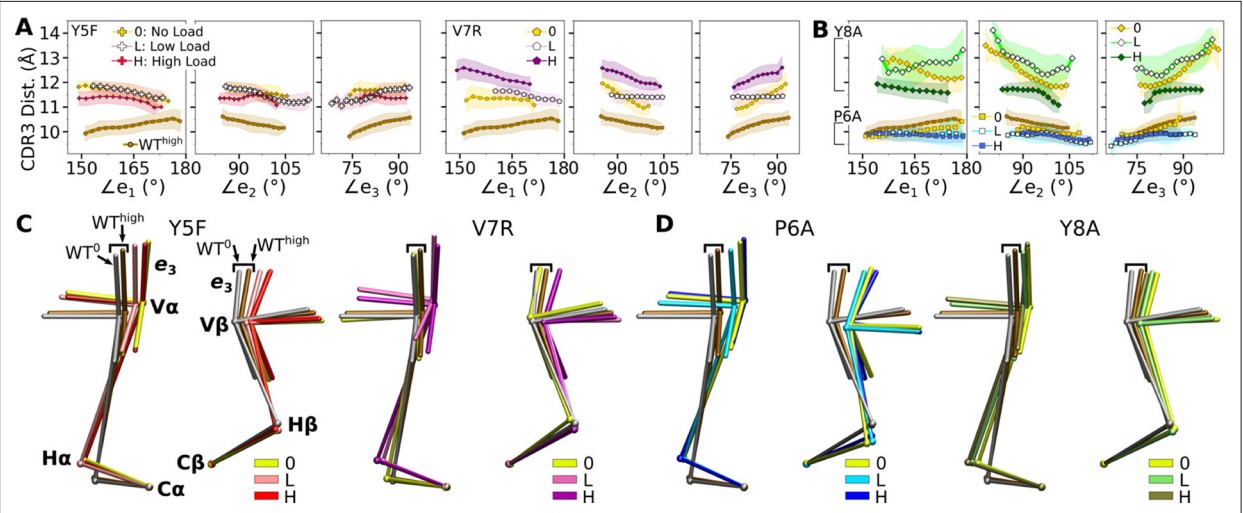

**Figure 7.** Mutant effects on the conformational dynamics of TCRαβ. (**A, B**) CDR3 distance versus triad arm angles for (**A**) modified agonists and (**B**) antagonists. Plot for WT^high in *Figure 4D* is reproduced for comparison. (**C, D**) Average BOCs of labeled complexes oriented to the constant domains of WT^high (V-C BOC and PCA) for (**C**) modified agonists and (**D**) antagonists. Average BOCs for WT^0 and WT^high are displayed for comparison (marked by angular brackets).

The online version of this article includes the following figure supplement(s) for figure 7:

**Figure supplement 1.** Vα-Vβ motion of mutant systems.

**Figure supplement 2.** Distribution of triad arm angles in mutant systems.

**Figure supplement 3.** Comparison of mutant average V-C BOCs and interfaces with those of WT^high.

**Figure supplement 4.** V-C motion of mutants.

The modified agonists had more Vα-Vβ contacts than WT^high while the antagonists had fewer, except for Y8A^high (*Figure 7—figure supplement 1A*). While the amplitude of Vα-Vβ motion was generally in a range similar to the WT systems (*Figure 4—figure supplement 1A* vs. *Figure 7—figure supplement 1B*), the CDR3 distance was larger for all mutant systems except for P6A, which had a weak dependence on triad angles (*Figure 7A, B*). The angles in turn varied among systems and loading conditions (*Figure 7—figure supplement 2*). These results suggest that point mutations to the peptide cause alterations in the load-dependence of the interface and the Vα-Vβ motion.

The mutants affected the average V-C BOC similarly as that for dFG^high (*Figure 7C and D* vs. *Appendix 1—figure 1C*). Among them, Y8A^high had an average BOC approaching that of WT^high, which aligns with the comparable number of contacts with pMHC (*Figure 6A*). However, the location of its Hα differed (*Figure 7D*), and the CDR3 distance was larger (*Figure 7B*). To quantify deformation of the average BOC, we measured displacements of centroids from the corresponding ones in WT^high, which were overall greater for the α chain than the β chain (*Figure 7—figure supplement 3A B*). Consistent with this, the mutants had fewer Vα-Cα contacts than WT^high and a similar number of Vβ-Cβ contacts (*Figure 7—figure supplement 3C D*).

Similar to the WT systems, the greater motion of the α chain than the β chain was observed in the mutant systems, as seen from the differences in V-C PC1 amplitudes (*Figure 7—figure supplement 4A B*). However, dot products of the BOC PC1 between WT and mutants revealed that the direction of motion differed by varying degrees, which was more for V7R^high and Y8A (*Figure 7—figure supplement 4C* vs. *Figure 5B*). Thus, point mutations on the WT peptide can affect the conformational motion of the whole TCRαβ, in addition to the average BOC.

To further test effects of point mutations, we introduced in silico point mutations P6A and Y8A to the WT complex (WT to antagonists) and conversely introduced A6P and A8Y mutations to the P6A and Y8A complexes, respectively (antagonists to WT). The in silico antagonists did exhibit reduction in contacts with pMHC while the results were mixed for the in silico WT, especially for A8Y where the introduced tyrosine is bulkier than the original alanine. Nevertheless, these tests support the above results based on the original crystal structures (See Appendix 2 for details).

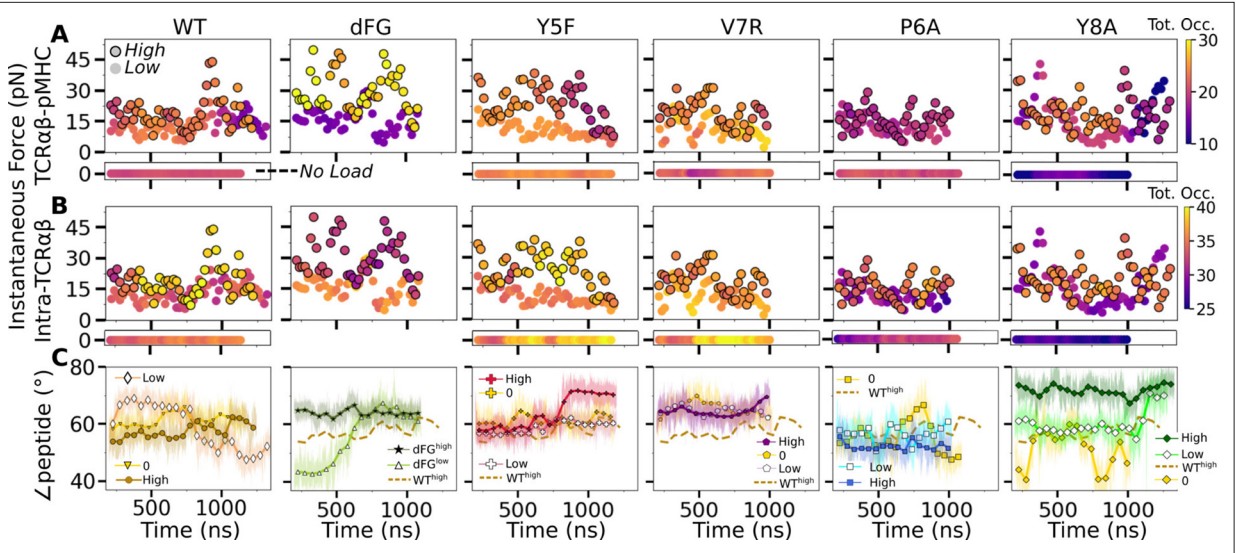

**Figure 8.** Relationship between force and interfacial behavior. (**A, B**) The total contact occupancy measured in 40-ns overlapping intervals starting from 200 ns (see Time-dependent behavior in Computational methods). (**A**) TCRαβ-pMHC (intermolecular) and (**B**) intra-TCRαβ (intramolecular) contacts excluding Cα-Cβ. Cases without load are shown as horizontal bars below each panel. Plots for low load systems (*Table 1*) do not have outlines. (**C**) Angle between antigenic peptide and the line between centroids of the triads for Vα and Vβ (Peptide and V-module angle). Thin lines: values at individual frames. Symbol: 50-ns running average.

The online version of this article includes the following figure supplement(s) for figure 8:

**Figure supplement 1.** Motion at the interface related to ∠peptide.

## Load- and time-dependent interfacial response

To probe the dynamic relation between the TCRαβ-pMHC (intermolecular) interface and intra-TCRαβ (intramolecular) interfaces formed between subdomains of the complex, we calculated the total occupancy of the high-occupancy contacts in respective cases (*Figure 8A, B*). For the intramolecular contacts, we excluded the Cα-Cβ interface contacts since they are larger in number compared to other interfaces and did not differ significantly across different systems. Thus, the C-module moves mostly as a single unit (*Hwang et al., 2020*).

For WT[0], the intermolecular contact occupancy stayed at around 20 (WT in *Figure 8A*, horizontal bar on the bottom) and for WT[low], it decreased later in simulation (WT in *Figure 8A*, darkening of circles without outline). In comparison, the intramolecular contact occupancy remained relatively constant for both WT[0] and WT[low] (WT in *Figure 8B*, horizontal bar on the bottom and circles without outline). For WT[high], the intermolecular contact occupancy was steady even with wider fluctuation in force (WT in *Figure 8A*, outlined circles), and the intramolecular occupancy also remained high, indicating the subdomains are held together tightly (WT in *Figure 8B*, outlined circles). For dFG[low], the intermolecular contact occupancy stayed low and intramolecular occupancy was relatively constant (dFG in *Figure 8A and B*, circles without outline). In dFG[high], the contact occupancy with pMHC increased (dFG in *Figure 8A*, outlined circles), but the intramolecular contact occupancy became low (dFG in *Figure 8B*, outlined circles), which suggests that the complex is not as tightly coupled compared to WT.

For modified agonists, the no load and low load cases had overall higher occupancy, both with pMHC and within TCRαβ, but occupancy fluctuated more as can be seen by the changes in colors in the occupancy trajectories (Y5F and V7R in *Figure 8A and B*, horizontal bars on the bottom and circles without outlines). Under high load, intermolecular contact occupancy decreased over time (Y5F and V7R in *Figure 8A*, darkening of outlined circles) while intramolecular contact occupancy either increased (Y5F[high]) or decreased (V7R[high]) relative to the respective low load cases. For antagonists, both occupancy measures were lower than the WT, and further reduction could be seen over time in some cases (P6A and Y8A in *Figure 8A and B*, darkening of colors in outlined circles).

The stability of the TCRαβ-pMHC interface also manifested into their relative motion, which was quantified by the angle between the least-square fit line across the backbone Cα atoms of the antigenic

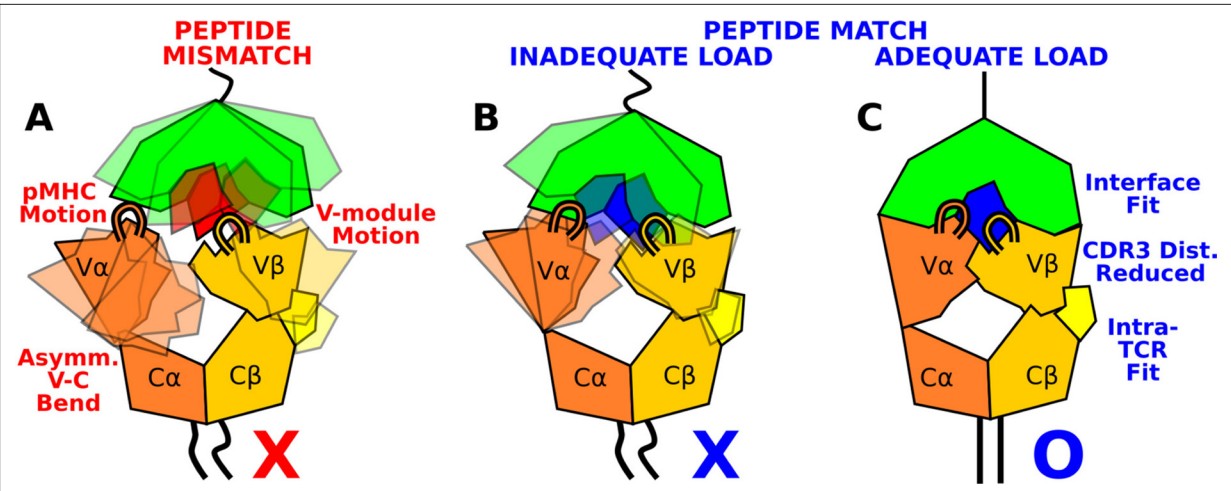

**Figure 9.** Model for peptide screening. (**A**) Non-matching pMHC or (**B**) matching pMHC but without adequate load do not stabilize the asymmetric V-C framework motion that affects the interfacial stability as measured by the CDR3 distance (CDR3 loops are shown above the V-module). (**C**) Matching pMHC with adequate load results in an overall tighter fit.

peptide and the unit vector formed between the centroids of Vα and Vβ (*Figure 8—figure supplement 1A–D*). For WT, the peptide angle fluctuated more for WT^low than WT^high (WT in *Figure 8C*) where 58.4°±3.6° (avg±std after 500 ns) for WT^high reflects a diagonal binding. This is consistent with the greater transverse RMSF of WT^low mentioned earlier which may assist with destabilizing the interface. For dFG^low, the peptide changed orientation by more than 20°, and for dFG^high, it stabilized, but at a higher value than WT^high, which also was reached in dFG^low later during simulation, suggesting a more orthogonal binding (dFG in *Figure 8C*). For modified agonists, similar to the behaviors of the total intra- and intermolecular contact occupancy, the peptide angle was affected more under high loads, again becoming more orthogonal compared to WT^high (Y5F and V7R in *Figure 8C*). For antagonists, the angle overall fluctuated more under no load or settled to different values under high load. Since the antagonists are loosely coupled (low occupancy in *Figure 8A and B*), settling of the angle does not indicate stabilization of the interface, as evident from the positional shift of the α2 helix of V7R^high or Y8A^high (*Figure 8—figure supplement 1G H*) compared to WT^high (*Figure 8—figure supplement 1E*).

## Discussion

The present study elucidates how the load-dependent TCRαβ framework motion influences the dynamics of the TCRαβ-pMHC interface (*Figure 9*). Instead of using defined conformational changes as seen in other catch bond systems, TCRαβ activates the catch bond via altering its conformational dynamics. A main feature of the TCRαβ framework is the smaller number of contacts for the Vα-Cα compared to the Vβ-Cβ interface. This causes an asymmetric V-C motion, primarily bending, where Vα moves more compared to Vβ relative to the C-module, which serves as a base. This in turn generates relative motion between Vα and Vβ, which can destabilize the contacts with pMHC, especially by affecting the distance between CDR3 loops that play the most direct role for sensing the bound peptide (*Figure 9A and B*). Applying a physiological level force stabilizes the interface by straining the whole complex into a more tightly coupled state, as can be seen by the increase of both inter- and intramolecular contacts in WT^high (*Figures 8 and 9C*). This physically plausible mechanism is based on collective analyses of all simulations in this study (*Figure 1*).

The CDR3 distance of WT^high (10.3±0.3 Å; *Figure 4—figure supplement 2C*) was shorter than that of WT^0 or WT^low (*Figure 4D*), and it is also shorter than the 10.9 Å CDR3 distance in the crystal structure (PDB 1AO7). The applied load slightly increases the spacing between pMHC and TCRαβ, which provides room for the CDR3 loops to adjust as well as allow other contacts to 'lock' into more stable states with higher and more persistent occupancy. Absence of load or low load do not properly

channel the framework motion and thereby increase exposure to water (*Figure 3F and G*) and destabilize the interface.

The Cβ FG-loop stabilizes the Vβ-Cβ interface, thereby contributing to the asymmetric V-C motion. It also controls the relative orientation between Vβ and Cβ, hence it affects the orientation of the CDR loops of the V-module with respect to the loading direction (*Appendix 1—figure 1C*). The behaviors of the dFG mutant are consistent with the reduced catch bond response observed experimentally (*Das et al., 2015*). Consistency in these findings between the present study and our previous simulations using JM22 TCR (*Hwang et al., 2020*) also underscores that the proposed mechanism based on the asymmetric framework motion is applicable to other TCRαβ systems. Furthermore, while the γδ TCR involved in transitional immunity operates as a slip-bond receptor, the Vγδ-Cαβ chimera forms a catch bond with its sulfatide ligand-loaded CD1d molecule (*Mallis et al., 2021*), which provides additional experimental support for the proposed mechanism.

In the absence of bond dissociation events within the microsecond-long simulation, the catch bond manifests as stabilization of high-occupancy contacts and interfacial fit under load. Conversely, slip bond will exhibit destabilization of the interface under load, as seen for antagonists (*Figure 8*). In the case of a higher affinity slip bond, the interfacial behavior could be insensitive to the applied load within the finite simulation time. In addition to agreeing with previous experiments mentioned above, our atomistic simulations can be used to design mutant TCRs possessing altered load dependence. For example, the V7-βG100 contact is present mainly in the high-load case (*Figure 3C*). The V7R peptide is a modified agonist as R7 forms contacts with residues other than βG100, albeit with lower occupancy (*Figure 6—figure supplement 2*). Point mutations of βG100 may lead to different behaviors depending on the type and size of the mutated residue. Another possibility is placing a disulfide bond to limit the Vα-Cα motion or to alter the relative V-C orientation. Extensive simulations are needed to more accurately predict behaviors of such mutants.

After engagement with a cognate pMHC under load, reversible transition to an extended state is possible. This has been observed both during in vitro single-molecule experiments using TCRαβ and on cells displaying the full αβTCR holoreceptor (*Das et al., 2015*; *Banik et al., 2021*). Since Vαβ-pMHC lacking the C-module forms a more stable binding (*Figure 3*) that was also observed in our previous simulations of the JM22 TCR (*Hwang et al., 2020*), the C-module likely undergoes partial unfolding in the extended state. Thus, while the folded C-module serves as the base for the asymmetric V-C motion screening for the matching pMHC, once a match is found, the reversible transitioning propelled by the partial unfolding of the C-module may agitate the membrane and activate the cytoplasmic domains of the surrounding CD3 subunits to initiate downstream signaling (*Reinherz et al., 2023*). A circumstantial evidence for the capacity of the C-module to unfold is that the Cα domain as well as parts of Cβ are occasionally unresolved in crystal structures, as in PDB 1AO7. Catch bond formation and subsequent reversible structural transitioning under applied load indicate that pMHC recognition requires energy input, for example from the actomyosin machinery. Further studies are needed to understand the energetics involved in pMHC recognition, signaling initiation, and ultimately T-cell activation.

In addition to TCRαβ, MHC may also respond to load. *Wu et al., 2019* suggested a partial separation of the MHCα1-α2 peptide-binding platform from β2m with the attendant lengthening of pMHC contributing to a longer bond lifetime. *Banik et al., 2021* observed a catch bond for CAR-pMHC, where just MHC is being pulled with an antibody. While we did not find a clear load or peptide-dependence in contacts between subdomains of MHC, since the entire TCRαβ-pMHC complex is under load, conformational changes in pMHC may contribute to the extended state of the complex. Yet, for T-cell-based cancer immunotherapy, mechanistic knowledge of the mechanosensing through a TCR has a greater practical significance (*Reinherz et al., 2023*).

A recent study using a laminar flow chamber assay fit the measured bead survival distribution using Bell's equation to estimate the zero-force off rate $k_{\text{off}}$ and the force sensitivity distance $x_\beta$ (*Pettmann et al., 2023*). They found a negative correlation between $k_{\text{off}}$ and $x_\beta$, to conclude that mechanical forces impair antigen discrimination. However, the force range tested was up to 100 pN, where even systems exhibiting catch bond in the 10–20-pN range will switch to a slip bond behavior. A catch bond exhibits a non-monotonic force versus bond lifetime profile, so that fitting with Bell's equation, an exponential function, leads to results that do not have a clear physical meaning. For example, $x_\beta$ in *Pettmann et al., 2023* was less than 1 Å in magnitude in all systems, which is shorter

than the length of a single covalent bond. They also performed steered MD simulation that applies hundreds of pN forces, which is inadequate for studying behaviors of the system under loads in the 10–20-pN range (*Hwang et al., 2020*). Use of a coarse grained model without appropriately incorporating atomistic properties of the TCR further makes it difficult to compare their simulation with wet laboratoryexperiments.

We earlier proposed that the residues of the antigenic peptide play a role more as 'teeth of a key' for screening the TCRαβ-pMHC interaction fitness rather than bearing applied loads (*Hwang et al., 2020*; *Reinherz et al., 2023*; *Figure 1*, 'bittings of a key'). The present study confirms this notion through simulations of mutant systems, where several contacts across the interface with pMHC were impaired due to a single-residue mutation on the peptide in ways that reflect the functional outcome of the mutation. In considering how a T-cell may respond to an unknown peptide, the pMHC motion and the asymmetric V-C motion are two points of guidance (*Figure 9*). Stabilization of the inter- and intra-molecular interfaces throughout the whole complex under 10–20-pN load would indicate a cognate TCRαβ-pMHC interaction (*Figure 8*). Since these features are based on overall TCRαβ-pMHC complex dynamics, rather than changes to specific contacts or a particular conformational change, they can be used to predict fitness of other TCRαβ-pMHC combinations. Such tests involve performing many all-atom MD simulations and trajectory analyses. An in silico method would be needed that efficiently predicts dynamic properties of the complex based on sequence and structural data only. Atomistic insights gained from the present study will be helpful for developing such a method in future studies.

## Materials and methods
### Structure preparation
Structure preparation was done using CHARMM (*Brooks et al., 2009*). Simulation systems were based on PDB 1AO7 (*Garboczi et al., 1996a*) 1QSE, 1QRN, and 1QSF (*Ding et al., 1999*) and 3QFJ (*Scott et al., 2011*). Residues from the TCR α- and β-chains were renumbered sequentially from the original non-sequential numbering in the PDB. Throughout the paper we use the renumbered index to refer to a residue. Residues differing at a few locations in some of the PDB files were converted so that all systems have identical sequences except for point mutations introduced in the Tax peptide (details are given below). Disulfide bonds between cysteine residues were introduced as noted in the PDB file. Histidine protonation sites were determined based on the 1QSE crystal structure to promote hydrogen bond formation with neighboring residues. Where neighboring residues were unlikely to hydrogen bond, we assigned the water-facing nitrogen of histidine as charged. This led to protonation of the $N^\delta$ atom for all histidine residues except for MHC H263 and β2m H84, where the $N^\epsilon$ atom was protonated. For truncated structures, crystal waters within 2.8 Å from the protein atoms were kept in the initially built system. For full structures, all crystal waters were kept.

We extended the termini of the TCRαβ-pMHC complex as handles for applying positional restraints (*Figure 2*, "added strands") (*Hwang et al., 2020*). For MHC, we used the sequence from UniProt P01892, where [276]LSSQPTIPI[284] was added after E275. For TCRαβ, sequences for the added strands were from GenBank ABB89050.1 (TCRα) and AAC08953.1 (TCRβ), which were [201]PESSCDVKLVEKS-FETDT[218] and [246]CGFTSESYQQGVLSA[260], respectively. After adding the strands, a series of energy minimization and a short MD simulation in the FACTS implicit solvent environment (*Haberthür and Caflisch, 2008*) were performed to relax them and bring together the C-terminal ends of the two TCR chains. The first two N-terminal residues of TCRβ were missing in all structures except for 3QFJ, so they were added and briefly energy minimized.

1AO7 (Tax peptide): In the original PDB 1AO7, coordinates for the Cα domain (D116–S204) and parts of Cβ (E130–T143, K179–R188, S219–R228) are missing. The coordinates listed are based on the renumbered indices. These were built using PDB 1QSE. For the Cα domain, we aligned the Vα domain of 1AO7 and 1QSE (K1–P115) based on their backbone $C_\alpha$ atoms and added the missing Cα domain residue coordinates to 1AO7. After this, we performed a brief energy minimization on the added domain while fixing positions of all other atoms of 1AO7. For missing residues in the Cβ domain, we used backbone $C_\alpha$ atoms of two residues each before and after the missing part to align 1QSE to 1AO7 and filled in coordinates, followed by a brief energy minimization of the added part in 1AO7. In this way, the TCRαβ-pMHC interface of the original 1AO7 is preserved. By comparison, previous simulations mutated PDB 1QRN back to WT (*Ayres et al., 2016*), which corresponds to the A6P in

silico WT system (Appendix 2), or converted a high-affinity variant of A6 (PDB 4FTV) by mutating β-chain residues, in particular nearly the entire CDR3 loop (*Rangarajan et al., 2018*). Compared to our approach, those preparation methods thereby introduce more perturbation to the interface with pMHC.

The β2m residues C67 and C91 were reverted (C67Y, C91K) based on UniProt P61769 referenced in PDB 1AO7. These agree with the β2m sequence in other structures.

1QRN (P6A): Except for the two N-terminal residues of TCRβ, there were no missing coordinates. This also applies to 1QSE and 1QSF. The following conversions were made to match the sequence with other structures: K150S (TCRα), and A133E and E134A (TCRβ).

1QSE (V7R): No residue conversion was made.

1QSF (Y8A): The following conversions were made: A219R (MHC) and A225T (TCRβ).

3QFJ (Y5F): There were no missing residues. We made the D204N conversion in TCRβ.

WT truncated complexes: For truncation, we used the constructed 1AO7 complex.

- Vαβ: the last residues were αD111 and βE116.
- Tαβ: the last residues were αD206 and βG247 (no C-terminal strands).
- Vαβ-pMHC: includes Vαβ, peptide, β2m, and MHC. The last residue of MHC was L276.
- $WT^0$: WT complex without the added C-terminal strands, as for Tαβ.
- dFG: residues βL218–βP231 removed from the corresponding WT complex. βG217 and βV232 were covalently joined.

## MD simulation protocol

### Solvation and equilibration of simulated systems

We used CHARMM (*Brooks et al., 2009*) to prepare simulation systems before the production run. The solvation boxes were orthorhombic for systems with pMHC and cubic for those without pMHC. For TCRαβ-pMHC, the size of the initial water box was such that protein atoms were at least 12 Å away from the nearest transverse face of the box and 25 Å from each longitudinal face. The extra space in the longitudinal direction was to initially test and select extensions of the complex for longer simulations in *Table 1*. For solvation, we used the TIP3P water. Water molecules with their oxygen atoms less than 2.8 Å from protein heavy atoms were removed. Neutralization of the system was done using $Na^+$ and $Cl^-$ ions at about 50 mM concentration. Crystal water molecules were kept in this procedure.

After neutralization, a five-stage energy minimization was applied where protein backbone and side chain heavy atoms were progressively relaxed (*Hwang et al., 2020*). This was followed by heating from 30 K to 300 K for 100 ps and equilibration at 300 K for 200 ps. Backbone heavy atoms were positionally restrained with 5-kcal/[mol·Å$^2$] harmonic spring constant during heating and equilibration, except for structures involving 1AO7 that originally had more missing residues, where a 2-kcal/[mol·Å$^2$] restraint was used. We then performed a 2 ns CPT (constant pressure and temperature) simulation at 1 atm and 300 K. We applied a 0.001-kcal/[mol·Å$^2$] restraint on backbone $C_\alpha$ atoms. The CHARMM DOMDEC module (*Hynninen and Crowley, 2014*) was used to parallelize the simulation. We applied the SHAKE method to fix the length of covalent bonds involving hydrogen atoms, and used a 2-fs integration time step.

### Production runs

Production runs were performed using OpenMM (*Eastman et al., 2017*). We used the CHARMM param36 all-atom force field (*Huang and MacKerell, 2013*) and the particle-mesh Ewald method to calculate electrostatic interactions. We used an Ewald error tolerance of $10^{-5}$ which is 1/50 of the default value in OpenMM, for accuracy. The cutoff distance for nonbonded interactions was 12 Å, and the Nose-Hoover integrator of OpenMM at 300 K was used, with a 2-fs integration time step. We ran OpenMM on GPUs with mixed floating point precision. Below are specific steps of the MD protocol relevant to individual systems in *Table 1* and *Table 2*.

### TCRαβ-pMHC with load

#### Laddered extension with added strands

To apply load, $C_\alpha$ atoms of the C-terminal ends of the added strands in the complex (*Figure 2A*, blue spheres) were held by 1-kcal/[mol.Å$^2$] harmonic positional restraint at a given extension during the

simulation. Restraints were applied to the $C_\alpha$ atom of MHC I284 and to the center of mass of two $C_\alpha$ atoms: αT218 and βA260. A flat-bottom distance restraint was applied to the latter two atoms to prevent large separation. It was activated when the distance of the two $C_\alpha$ atoms was greater than 10 Å, where a 1.0-kcal/[mol.Å²] harmonic potential was applied. Starting with the initially built complex, we performed a 4-ns run then increased the extension by shifting centers of the positional restraints on terminal atoms by 2 Å at each end, for a total 4 Å added at each extension, for the next 4-ns run. The process continued to yield 4–6 extensions.

After each extension run, we truncated the water box such that the length of the box was 12 Å larger than the maximum span of the complex on each side, and re-neutralized the system. A representative water box size is 218×97×90 Å³ for WT$^{high}$, containing 187,250 atoms. Since the system was already equilibrated from the previous run, we used a simpler energy minimization scheme where backbone and side chain heavy atoms were restrained by 10-kcal/[mol·Å²] and 5-kcal/[mol·Å²] harmonic potentials, respectively, and 200 steps of steepest descent followed by 200 steps of adopted-basis Newton-Raphson energy minimization was performed. Heating, equilibration, and the initial 2-ns dynamic runs with positional restraints were carried out as explained above. We then carried out 60–100 ns production runs for each extension and selected two or three extensions to continue for longer than 1000 ns.

## Selecting extensions

We measured the average force on the complex during each 60–100-ns simulation, then selected two extensions where the average force generated was representative of a 'low' (around 10 pN) and 'high' (over 15 pN) load on the TCR. These values were based on the experimental 10–20-pN catch bond activation force range (*Das et al., 2015*; *Liu et al., 2016*).

In some cases, in particular at low extensions, the flexible added strand either folded onto itself or made contacts with the C-module of TCRαβ, effectively shortening the span of the complex. Factors such as this, together with differences in conformational behaviors of the complex, affected the average force for a given extension. Thus we had to test and choose among different extensions for each system. We also ran 1–2 replicate simulations of comparable length (1 μs) at given extensions except for systems involving dFG and in silico mutants. However, even with nearly the same extensions used, measured forces in replicate simulations varied due to the reasons explained above. Additionally, the average force based on the initial 60–100 ns and after 500 ns differed. For detailed analysis, we thus chose sets of runs where higher extension led to higher average force. However, in all runs, we found that the average load rather than extension correlates better with the behavior of the TCRαβ-pMHC interface, which underscores the consistency of the load dependence found in our analysis. The final selection and average forces are in *Table 1*.

## Other systems
### TCRαβ-pMHC without load

These systems include WT$^0$ and the no load complexes with point mutations to the Tax peptide. To prevent the complex from turning transversely in the elongated orthorhombic box, we applied a weak 0.2-kcal/[mol·Å²] harmonic positional restraint on select $C_\alpha$ atoms in the MHC α3 domain that had RMSF below about 0.5 Å in both WT$^{low}$ and WT$^{high}$, which were P185–T187, L201–Y209, F241–V247, and T258–H263.

### Vαβ-pMHC

We applied a 0.01-kcal/[mol·Å²] harmonic restraint to the backbone $C_\alpha$ atoms of the MHC α3 domain (residues P185-L276) to prevent the complex from turning transversely in the orthorhombic box. The restraints are 20 times weaker than those used for TCRαβ-pMHC complexes mentioned above. This was because Vαβ-pMHC is smaller in both size and aspect ratio.

### Vαβ, Tαβ, dFG

No positional restraints were applied. A representative system size is, for Tαβ, a 92.8 Å³ cubic water box containing 75,615 atoms.

### dFG-pMHC

The FG-loop deletion was done after initially preparing (solvation and neutralization) the WT complex in the extended water box. After deletion, the system was re-neutralized. Subsequently, laddered extension, selecting extensions for high and low load cases, and longer production runs were performed as explained above.

### In silico mutants

Each in silico mutation (*Appendix 2—table 1*) was performed for low and high load extensions of the complex. To use similar extensions as in the original complexes, we used the last frame of the 4-ns laddered extension simulation. After introducing the in silico mutation, we inspected the structure to ensure there was no steric clash with neighboring residues or water molecules. We performed a short energy minimization to relax the modified residue while keeping coordinates of all other residues except for residues immediately before and after the mutated one on the peptide. We then truncated the water box and re-neutralized the system, after which steps from the initial energy minimization up to the final production run followed the same procedure as explained above.

## Trajectory analysis

Coordinates were saved every 20 ps (0.02 ns) during production runs, resulting in 50,000 coordinate frames for 1000 ns. We excluded the initial 500 ns when calculating averages and standard deviations in the number of contacts, CDR3 distance, BSA, PCA values, and angle data. Since all systems were simulated for a minimum of 1 µs, this leaves at least 25,000 frames. We report data prior to 500 ns in trajectory plots and contact occupancy heat maps (e.g. *Figure 3B–E*).

### Calculating force

Force on a restrained atom or the center of mass of the C-terminal atoms of the added strands in TCRαβ was calculated based on the deviation of its average position from the center of the harmonic potential, multiplied by the spring constant used (*Hwang, 2007*; *Hwang et al., 2020*). Average force in *Table 1* was computed from 500 ns to the end of the simulation. Instantaneous force in *Figure 8A and B* was computed in 40-ns overlapping intervals starting from 200 ns, *i.e.*, 200–240 ns, 220–260 ns, 240–280 ns, etc.

### CDR3 distance

The CDR3 distance (e.g. *Figure 4—figure supplement 2A–C*) was measured using the midpoint between backbone $C_\alpha$ atoms of two residues at the base of each CDR3 loop. They were: T92 and K97 for CDR3α and R94 and E103 for CDR3β.

### Contact analysis

We used our previously developed method (*Hwang et al., 2020*). Briefly, H-bonds (including salt bridges) were identified with the 2.4 Å donor-acceptor distance cutoff. Nonpolar contacts were identified for atom pairs that are within 3.0 Å and both have partial charges less than $0.3e$ ($e = 1.6 \times 10^{-19}$ C) in magnitude. The average occupancy was measured as the fraction of frames over which a bond is present during the measurement period. Instantaneous occupancy was measured as a 40-frame (0.8-ns) rolling average. The average occupancy of a contact represents its abundance during the simulation period while the instantaneous occupancy represents its temporal intensity.

For counting the number of contacts (e.g. *Figure 3A*, *Figure 4A*, *Figure 5—figure supplement 1*), we used contacts with the average occupancy greater than 50% and at least an 80% maximum instantaneous occupancy after the initial 500 ns. Contact occupancy heat maps (e.g. *Figure 3C–E*) report those with the overall average occupancy greater than 30%, and the maximum instantaneous occupancy during the simulation greater than 80%.

The Hamming distance $\mathcal{H}$ (e.g. *Figure 3B*) was measured using contacts with greater than 80% average occupancy during the first 50 ns.

## BSA

For the BSA calculation (e.g. *Figure 3G*), we used residues in the V-module with the maximum instantaneous contact occupancy with pMHC greater than 80%. We calculated the surface area for the selected residue contacts and added them to get the total BSA. Per-residue BSA is the total BSA divided by the number of residues forming the contacts in the given time interval. The reported values (e.g. *Figure 3G*) are respective averages after 500 ns.

## Variable domain triads and PCA

Triads (orthonormal unit vectors) were constructed for Vα and Vβ by modifying the procedure in *Hwang et al., 2020* for the A6 V-module. We used the backbone $C_\alpha$ atoms of six residues from the central four β-strands that make up the stably folded β-sheet core of each variable domain: for Vα, S19-Y24, F32-Q37, Y70-I75, Y86-T91 and for Vβ, T20-Q25, S33-D38, F74-L79, V88-S93. The $C_\alpha$ atoms of these residues have RMSF in $WT^{high}$ near or less than 0.5 Å, and they correspond to two matching segments on each of the inner and outer β-sheets of the immunoglobulin fold. The center of mass of the $C_\alpha$ atoms of the selected residues was used for the centroid of each triad. The $\mathbf{e}_3$ arm of the triad was assigned along the major axis of the least-square fit plane of the selected atoms in each domain, which is parallel to the β-strands and points to the CDR3 loop (*Figure 4B*). The $\mathbf{e}_1$ arm was assigned by taking the direction from the center of masses of the selected atoms from the inner to the outer β-sheets of each variable domain and making it perpendicular to $\mathbf{e}_3$. The $\mathbf{e}_2$ arm was then determined as $\mathbf{e}_2 = \mathbf{e}_3 \times \mathbf{e}_1$.

PCA was performed on the trajectory of the two triads using a custom FORTRAN95 program (*Hwang et al., 2020*). The PC amplitude (e.g. *Figure 4—figure supplement 1A*) corresponds to the rotational motion of these arms in units of radians. The PC vector for the 6 arms of the two triads is an 18-dimensional unit vector. To compare directions of two PCs (e.g. *Figure 4—figure supplement 1C*), the absolute value of the dot product between them was calculated, which ranges between 0 and 1. To project the Vα-Vβ triad for a given frame to a PC direction (*Figure 4—figure supplement 1D*), the average triad calculated after the initial 500-ns was subtracted from the triad, then a dot product was formed with the PC vector.

## V-C BOC and PCA

The V-C BOC (*Figure 5A*) was assigned based on the method we developed previously (*Hwang et al., 2020*). For beads representing the V-module, centroids of the two triads were used. For the C-module, the center of mass of backbone $C_\alpha$ atoms of the following residues in each domain were used: for Cα, A118–R123, V132–D137, Y153–T158, S171–S176, and for Cβ, T143–A148, L158–N163, S192–V197, F209–Q214. We used αN114 for Hα, and for Hβ, the center of mass between βD117 and βL118 was used, which had large RMSF in $WT^{high}$.

We aligned coordinate frames for all simulations to the first frame of $WT^{high}$ based on atoms used to assign beads for the C-module. In this way, motion of the V-module relative to the C-module can be analyzed. Also, by using a common reference structure (first frame of $WT^{high}$), average BOCs can be compared, as in *Figure 7C and D*. PCA of the V-C BOC was performed using the 6 beads representing the centroids and hinges. PCA for the V-module triads was done separately. Since the reference of motion is the C-module, directions of PCs for the V-module triads indicate motion of the V-module relative to the C-module (arrows on triad arms in *Figure 5A*), which complements the direction of the V-module centroids obtained from PCA of the V-C BOC (arrows on centroids in *Figure 5A*).

## Time-dependent behavior

For the total occupancy in *Figure 8A and B*, we only considered contacts with greater than 50% overall occupancy and over 80% maximum instantaneous occupancy during the entire simulation period. In this way, changes in high-quality contacts under fluctuating force for a given trajectory can be monitored. For each 40-ns window, we calculated the average occupancy of selected contacts and added them to obtain the total occupancy. For intermolecular contacts, interfaces between MHC-Vα, MHC-Vβ, peptide-Vα, and peptide-Vβ were considered. For intramolecular contacts, Vα-Vβ, Vα-Cα, and Vβ-Cβ were considered.

### Peptide and V-module angle

For *Figure 8C*, at each coordinate frame we calculated the least-square fit line for the peptide backbone $C_\alpha$ atoms and calculated a dot product of its direction with a unit vector pointing from the centroid for the triad of Vα to that of Vβ.

## Acknowledgements

This work was funded by US NIH Grants P01AI143565 and R01AI136301. Simulations were performed by using computers at the Texas A&M High Performance Research Computing facility.

## Additional information

### Funding

| Funder | Grant reference number | Author |
|---|---|---|
| US National Institutes of Health | P01AI143565 | Robert J Mallis<br>Matthew J Lang<br>Ellis L Reinherz<br>Wonmuk Hwang |
| US National Institutes of Health | R01AI136301 | Matthew J Lang<br>Ellis L Reinherz |

The funders had no role in study design, data collection and interpretation, or the decision to submit the work for publication.

### Author contributions

Ana C Chang-Gonzalez, Conceptualization, Data curation, Formal analysis, Validation, Investigation, Visualization, Methodology, Writing – original draft, Writing – review and editing; Robert J Mallis, Conceptualization, Investigation, Writing – review and editing; Matthew J Lang, Ellis L Reinherz, Conceptualization, Funding acquisition, Investigation, Writing – review and editing; Wonmuk Hwang, Conceptualization, Formal analysis, Supervision, Funding acquisition, Validation, Investigation, Visualization, Methodology, Writing – review and editing

### Author ORCIDs

Ana C Chang-Gonzalez  https://orcid.org/0000-0002-1517-4172
Robert J Mallis  http://orcid.org/0000-0002-2087-9468
Matthew J Lang  http://orcid.org/0000-0002-8198-144X
Ellis L Reinherz  http://orcid.org/0000-0003-1048-5526
Wonmuk Hwang  http://orcid.org/0000-0001-7514-3186

### Decision letter and Author response

Decision letter https://doi.org/10.7554/eLife.91881.sa1
Author response https://doi.org/10.7554/eLife.91881.sa2

## Additional files

### Supplementary files
• MDAR checklist

### Data availability

The current manuscript is a computational study, so no experimental data have been generated for this manuscript. Sample analysis scripts are available on GitHub: https://github.com/hwm2746/a6tcr_anal_md/tree/main (copy archived at *Chang-Gonzalez and Hwang, 2024*).

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

## Appendix 1

### Allosteric effect of the Cβ FG-loop deletion

The higher number of Vβ-Cβ contacts (*Figure 5—figure supplement 1*) and smaller range of ∠TCRβ (*Figure 5D and E*) in the WT TCR suggests that the β chain mainly bears the load while the α chain adjusts to accommodate different loading conditions. Our previous experimental (*Das et al., 2015*) and computational (*Hwang et al., 2020*) studies showed that the Cβ FG-loop plays a critical allosteric role for the catch bond formation. To examine its role in A6 TCR, we performed simulations of dFG in isolation (*Table 2*) and under low and high loads (*Table 1*). Similar to the WT, the number of contacts with pMHC increased with load (*Appendix 1—figure 1A* and *Appendix 1—figure 2A and B*). The BSA for high-occupancy residues contacting pMHC was also greater for high load (*Appendix 1—figure 2C*). Thus, dFG may also possess a catch bond behavior, which agrees with experiment where a subdued catch bond was observed (*Das et al., 2015*). However, $\mathcal{H}$ increased early on and was slightly larger than that for WT$^{high}$ (*Appendix 1—figure 2*), and the CDR3 distance was higher compared to WT$^{high}$ (*Appendix 1—figure 1B*), which indicate an altered interface.

The conformation of the whole dFG was also affected. Relative to the C-module, the average BOC for the unliganded dFG was substantially different from those of dFG$^{low}$ and dFG$^{high}$, where the latter was similar to that of WT$^{high}$ (*Appendix 1—figure 1C*). In particular, Vβ of the unliganded dFG is more tilted, as there is a lack of support from the FG-loop (*Hwang et al., 2020*). When load is applied to the dFG-pMHC complex, dFG becomes less bent. Its tendency to return to the bent conformation would impose a strain on the interface with pMHC. This can be seen by the higher average load on dFG-pMHC complexes than WT-pMHC complexes under similar extensions (*Table 1*). Comparing between the amplitudes of PCs of α and β chains, a notable difference from the WT systems (*Figure 5C*) is that Hβ moves more than Hα for loaded dFG systems (*Appendix 1—figure 1D*, bottom row). Also, distributions of ∠TCRα and ∠TCRβ shift to larger and smaller values, respectively (*Appendix 1—figure 1E*). These indicate alterations in the conformation and motion of dFG. Furthermore, the CDR3 distance of dFG is elevated regardless of load or V-C angle (*Appendix 1—figure 1F*), suggesting a reduced allosteric control by the V-C motion.

The altered conformation of dFG causes the interface with pMHC to tilt as observed in our previous study of JM22, which is detrimental to the stability of the complex (*Hwang et al., 2020*). The increased motion of Hβ may also deliver more agitation to the interface with pMHC. Thus, even though full dissociation with pMHC was not observed within the simulation time, the dFG-pMHC complex is likely to be less stable compared to the WT complex.

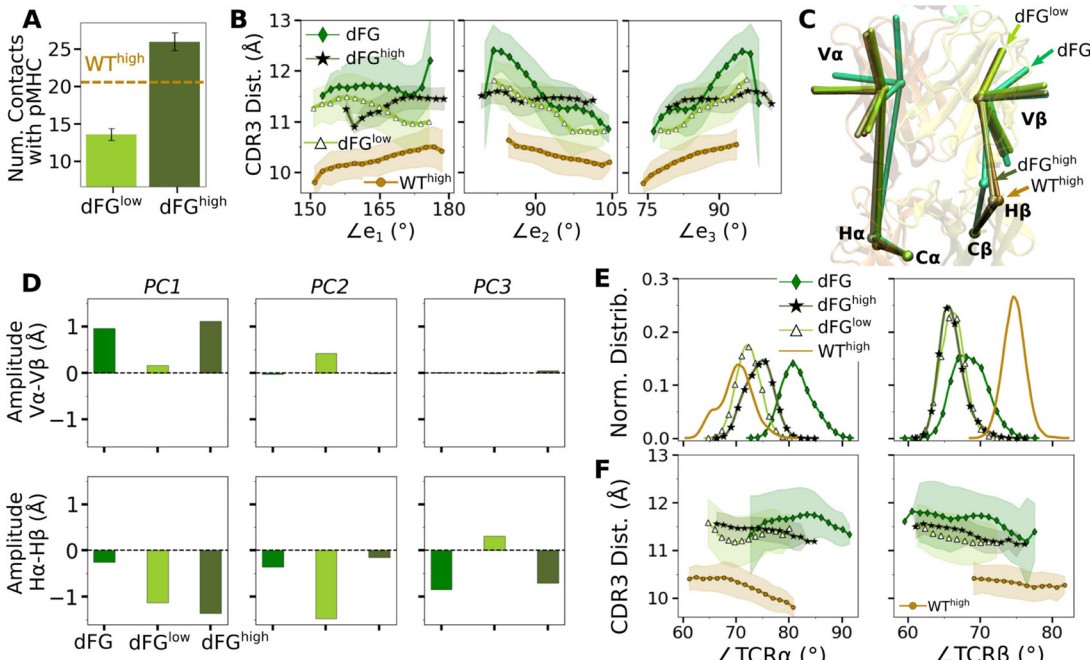

**Appendix 1—figure 1.** Effects of the Cβ FG-loop deletion. In all panels, the same criteria were used to measure values as for the WT systems in the corresponding figures. For comparison, respective data for WT[high] are shown. (**A**) Number of contacts with pMHC (*Figure 3A*). (**B**) CDR3 distance vs. the three Vα-Vβ triad arm angles (*Figure 4D*). (**C**) Average BOC of labeled complexes oriented to the C-module of WT[high]. The unliganded dFG has notably different average BOC (*Figure 7C and D*). (**D**) Differences in amplitudes between respective PC components of the α and β chains (*Figure 5C*). (**E**) Histogram of hinge angles and (**F**) CDR3 distance vs. hinge angles (*Figure 5D and E*).

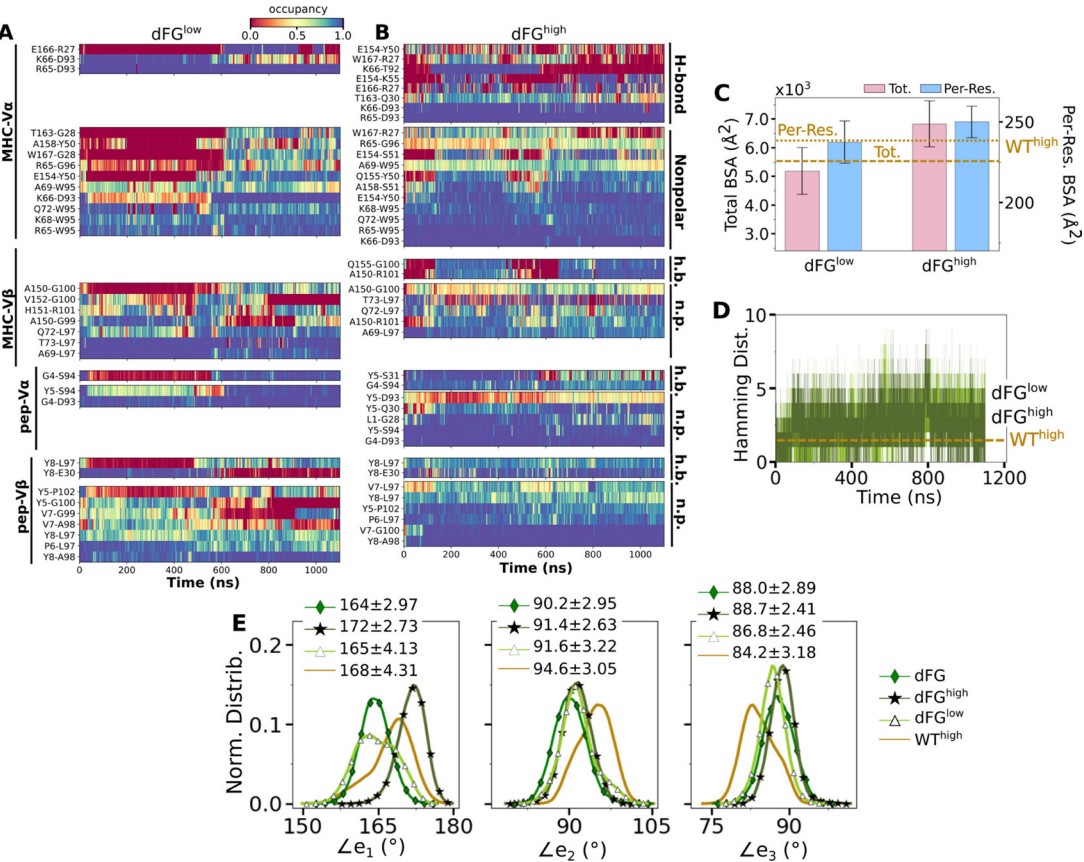

**Appendix 1—figure 2.** Effects of the Cβ FG-loop deletion on the interface with pMHC. The same criteria were used to plot as for the WT systems in the corresponding figures. (**A–B**) Contact occupancy heat maps (*Figure 3C and D*). (**C**) BSA (*Figure 3G*). (**D**) Hamming distance (*Figure 3B*). (**E**) Distribution of Vα-Vβ angles. Numbers are avg±std in respective cases (*Figure 4C*). In (**C–E**), data for WT$^{high}$ are shown as reference.

## Appendix 2

### In silico peptide mutants mimic the behaviors of target systems

We tested whether behaviors of different systems are interchangeable by making point mutations on peptides in silico, for which WT and antagonists were used (*Appendix 2—table 1*). For example, for $^{Y8A}WT^{high}$ (Y8A under high load switched to WT), we took Y8A$^{high}$ at the beginning of its production run and mutated A8 to Y8 (see In silico mutants in Computational methods). A main question is whether the in silico mutants attain behaviors of the switched systems during the finite simulation time. We found this to be the case although behaviors were not recapitulated perfectly. Compared to WT$^{high}$, the number of contacts with pMHC became lower for $^{WT}P6A$ (*Appendix 2—figure 1A*). Note that $^{WT}P6A^{low}$ had a higher force than $^{WT}P6A^{high}$ (24.7 vs 19.5 pN; *Appendix 2—table 1*). This was because the low extension in the former case allowed the loose C-terminal strands (*Figure 2A*) to form extensive nonpolar contacts with the C-module, especially with Cβ. This effectively shortened the length of the complex, which led to a higher average force as the extension was kept the same (Selecting extensions). Among the original systems, Y5F$^{high}$ had average force comparable to $^{WT}P6A^{low}$ (23.7 pN; *Table 1*), yet it had 1.8-fold more contacts, suggesting that the latter does behave like an antagonist (*Figure 6A* vs. *Appendix 2—figure 1A*). For in silico WT, the number of contacts with pMHC was comparable to that of WT$^{high}$ except for $^{Y8A}WT^{high}$ (*Appendix 2—figure 1A*). In the original Y8A$^{high}$, even though the number of contacts was at the level of WT$^{high}$ (*Figure 6A*), the smaller size of A8 caused CDR3β to extend. The altered interface can be seen by the initial rapid increase in $\mathcal{H}$ (*Figure 6—figure supplement 1D*). Mutating A8 to Y8 thereby forces the bulkier Y8 side chain to take an orientation different from that of WT. Thus, an in silico mutation of a residue to a comparable or smaller one is better tolerated than mutating to a bulkier one.

**Appendix 2—table 1.** Simulations of TCRαβ with in silico mutations on the peptide.
Load reported is average after 500 ns. Parentheses after the average load show standard deviation in forces measured in 40-ns intervals after 500 ns.

| PDB ID | Mutation | Extension (Å) | Time (ns) | Load (pN) | Label |
|---|---|---|---|---|---|
| 1AO7 | P6A | 182.9 | 1140 | 24.7 (13.9) | $^{WT}P6A^{low}$ |
| | | 187.5 | 1040 | 19.5 (5.67) | $^{WT}P6A^{high}$ |
| | Y8A | 182.3 | 1000 | 16.5 (5.02) | $^{WT}Y8A^{low}$ |
| | | 187.1 | 1000 | 28.6 (7.87) | $^{WT}Y8A^{high}$ |
| 1QRN | A6P | 175.2 | 1000 | 10.9 (6.09) | $^{P6A}WT^{low}$ |
| | | 186.2 | 1060 | 31.8 (7.57) | $^{P6A}WT^{high}$ |
| 1QSF | A8Y | 176.7 | 1000 | 10.0 (5.89) | $^{Y8A}WT^{low}$ |
| | | 187.4 | 1040 | 11.5 (5.85) | $^{Y8A}WT^{high}$ |

For $^{WT}Y8A$ and $^{P6A}WT$, a higher load led to more contacts with pMHC (*Appendix 2—figure 1A*). For $^{WT}Y8A^{high}$, the number was comparable to WT$^{high}$, which agrees with the case for the original Y8A$^{high}$ (*Figure 6A*). The BSA profiles of in silico mutants also followed a trend similar to the number of contacts with pMHC, which was lower for antagonists and higher for WT (*Appendix 2—figure 2A*). Differences in binding with pMHC can also be seen in the positional distribution of high-occupancy contacts, where $^{P6A}WT$ had relatively compact and evenly distributed contacts (*Appendix 2—figure 2B*), although not as extensive as Vαβ-pMHC or WT$^{high}$ (*Figure 3F*).

Regarding the Vα-Vβ interface, there were overall less contacts in the in silico antagonists than in silico WT (*Appendix 2—figure 2C*). However, a higher load did not result in greater number of Vα-Vβ contacts. The CDR3 distance was higher for in silico antagonists (*Appendix 2—figure 1B*, top), while for in silico WT their range became narrower, to 11–12 Å, which is similar to that of the modified agonist Y5F (*Figure 7A* vs. *Appendix 2—figure 1B*, bottom row). The CDR3 distance also stabilized over time in all in silico WT, which was less so for in silico antagonists (*Appendix 2—figure 2D*). However, similarly as the number of Vα-Vβ contacts, there was no consistent load-dependence between the CDR3 distance and triad arm angles. On the other hand, there was a stronger load dependence in the average V-C BOC. The in silico antagonists that were built based on WT bent towards those of the corresponding antagonists, though the extent was not large (*Figure 7D* vs. *Appendix 2—figure 1C*). The in silico WT in low load had average BOCs similar to those of the

original antagonists whereas average BOCs of high-load in silico WT approached those of the actual WT (*Appendix 2—figure 1D*). For $^{Y8A}$WT, this happened even though the forces experienced at the two extensions were only marginally different (10.0 vs 11.5 pN; *Appendix 2—table 1*). The Vα-Cα and Vβ-Cβ contacts were respectively lower for in silico WT than in silico antagonists, suggesting effects of the in silico mutations of the peptide did not propagate sufficiently to the whole TCRαβ during the simulation time (*Appendix 2—figure 2E*). The lower number of V-C contacts of in silico WT would have made it easier to unbend under higher load or extension.

The above results suggest that the in silico mutants behave like the target system to varying extents. This is likely because the rearranged interface between the V-module and pMHC of the base system cannot immediately be adjusted upon in silico mutation in loaded states.

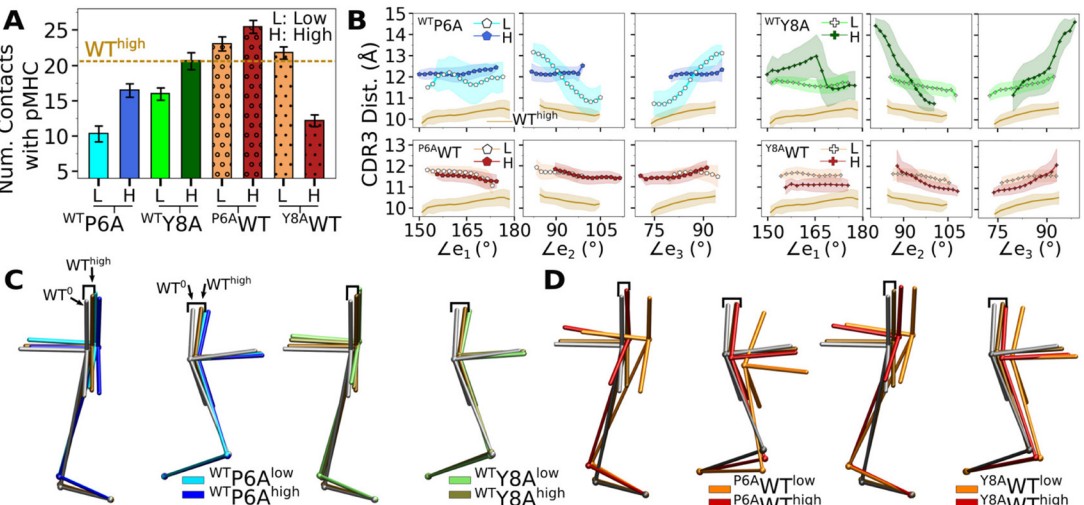

**Appendix 2—figure 1.** Simulations of in silico peptide mutants bound to A6. (**A**) Number of contacts with pMHC. Counts were made in the same way as in *Figure 3A*. Dashed line: average value for WT$^{high}$. Bars: std. (**B**) CDR3 distance vs. triad arm angles (*Figure 4D* and *Figure 7A and B*). (**C, D**) Average BOCs of (**C**) in silico antagonists, and (**D**) in silico WT. Average BOCs of WT$^0$ and WT$^{high}$ are shown as reference, marked by angular brackets (*Figure 7C and D*).

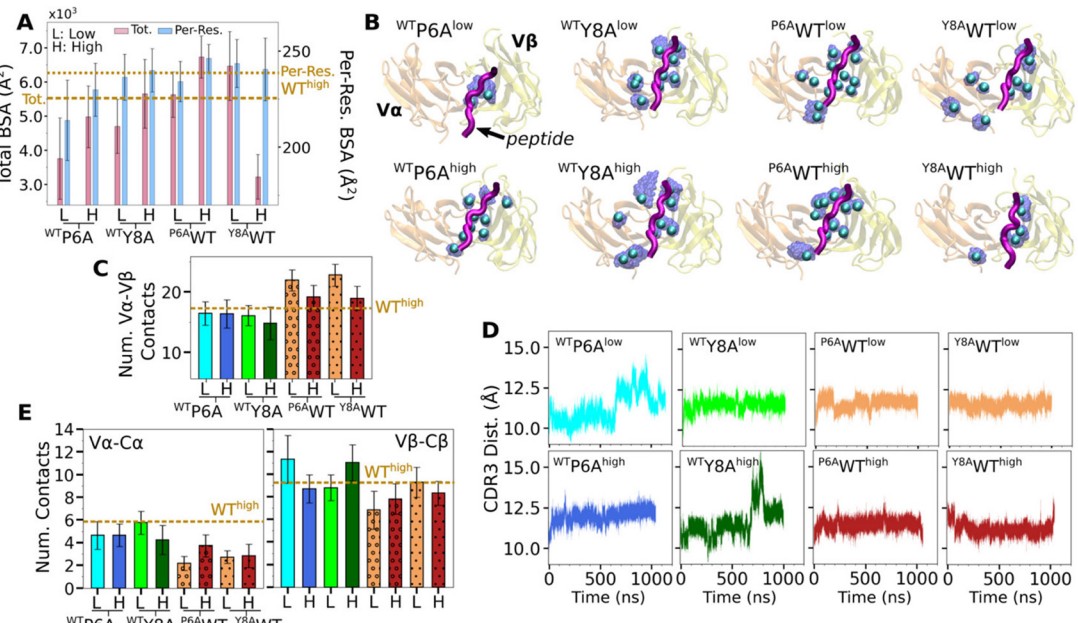

**Appendix 2—figure 2.** Behaviors of in silico mutants. (**A**) BSA (*Figure 3G*). (**B**) Positions of backbone Cα atoms of high contact occupancy residues (*Figure 3F*). (**C**) Vα-Vβ contact count (*Figure 4A*). (**D**) CDR3 distance trajectory (*Figure 4—figure supplement 2A–C*). (**E**) Vα-Cα and Vβ-Cβ contact counts (*Figure 5—figure supplement 1*).

## Appendix 3

### Relationship between the average and standard deviation of force

In *Table 1*, standard deviation (std) of force spans a wide range, from 2.46 pN (Y5F$^{low}$) to 11.8 pN (dFG$^{high}$). When it is plotted against the average force, a near-linear relationship can be seen (*Appendix 3—figure 1*). Thermodynamically, the force and position of the restraint (blue spheres in *Figure 2A*) form a pair of generalized force and the corresponding spatial variable in equilibrium at temperature 300 K, which is akin to the pressure $P$ and volume $V$ of an ideal gas. If $V$ is fixed, $P$ fluctuates. Denoting the average and std of pressure as $\langle P \rangle$ and as $\Delta P$ respectively, it has been shown that $\Delta P / \langle P \rangle$ is a constant (*Burgess, 1973*). In the case of the TCRαβ-pMHC system, although individual atoms are not ideal gases, since their motion lead to the force fluctuation of the restraints, the situation is analogous to the case of an ideal gas where pressure arises from individual molecules hitting the confining wall as the restraint. Thus, the near-linear behavior is a consequence of the system being many-bodied and at constant temperature. The linearity is also an indirect indicator that sampling of force in our simulation was reasonable.

In addition to the thermodynamic aspect, system-specific aspects influence the std of force. In *Appendix 3—figure 1*, Y8A$^{low}$ is an antagonist that had the smallest number of contacts with pMHC except for Y8A$^{0}$ without load (*Figure 6A*). dFG$^{low}$ also had similarly small number of contacts with pMHC (Appendix 1—figure 1A). The weakly held interface likely caused a wider conformational motion, leading to greater fluctuation in force relative to the average (dFG$^{low}$ and Y8A$^{low}$ in *Figure 8A*, symbols without outline).

Above results suggest that the fluctuation of force per se has no direct relation to the catch vs. slip bond mechanisms, although a comparatively larger std is indicative of potential instability.

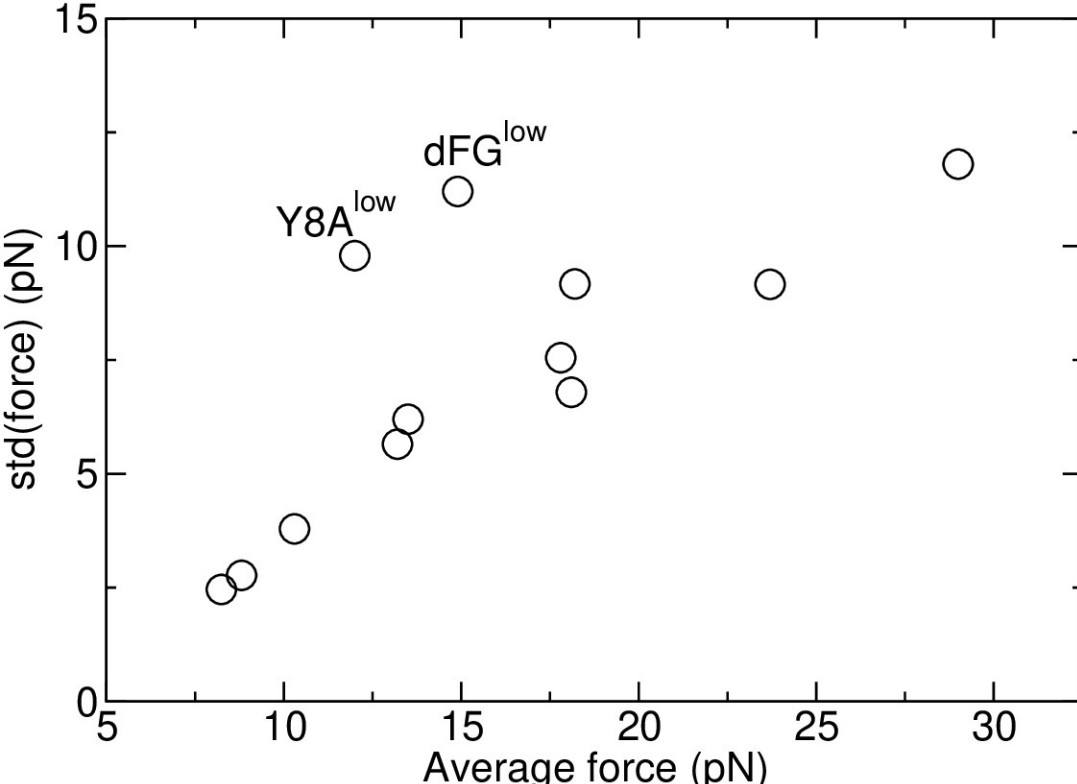

**Appendix 3—figure 1.** Standard deviation vs. average force in *Table 1*. Two major outliers are marked, that have the lowest number of TCRαβ-pMHC interfacial contacts among all loaded systems.

