## [Editor Report]

Using extensive atomistic molecular dynamics simulations, the authors analyzed the TCR/pMHC interface with different peptide sequences and protein constructs. The results provide important insights into the catch-bond phenomenon in the context of T-cell activation. In particular, the analysis points to convincing evidence that supports the role of force in further discriminating different peptides during the activation process beyond structural considerations.

---

## [Decision Letter]

**Decision letter after peer review:**

Thank you for submitting your article "Asymmetric framework motion of TCRαβ controls load-dependent peptide discrimination" for consideration by *eLife*. Your article has been reviewed by 3 peer reviewers, and the evaluation has been overseen by a Reviewing Editor and Qiang Cui as the Senior Editor. The reviewers have opted to remain anonymous.

Essential revisions:

1) From a technical point of view, discuss carefully the choice of using a restraining approach vs. applying a steady force, and clarify which better mimics the realistic situation. In addition, the principal component analysis can be done differently to ensure the most meaningful comparison between different cases.

2) In terms of results, discuss more explicitly the specific features that discriminate between catch-bond and slip-bond regimes. It is also valuable to explicitly suggest a set of experimentally testable predictions from the simulation study.

*Reviewer #1 (Recommendations for the authors):*

1. I suggest that the authors discuss why they chose to restrain the TCR/MHC separation, rather than devise an algorithm to apply a steady force. The issue with the restrained distance is that the forces reported for the different mutants are quite variable, and one might even say, not consistent.

2. Uncertainty should be reported as the forces in some form, or the magnitude of their fluctuation.

3. (Table1) it might be interesting to check whether the integral of the force over the three extensions reported in the table correlates with the TCR/pMHC binding strength, if these data are available.

4. A few simulation observations either appear speculative or are not well illustrated, (e.g. on p5), "short distance between restraints … allows wider transverse motion that in turn generates a shear stress or a bending moment at the interface". Given the complexity of this large biomolecular complex and its dynamics, I suggest making a greater effort to distinguish between what is actually observed and what the implications might be.

5. While there are various analyses of the simulation data, it would strengthen the paper greatly if the authors could provide specific experimentally testable hypotheses, eg., in the form of predicted responses to a mutant peptide, or mutations to the variable chains that could alter the fluctuations (e.g. disulfide crosslinking).

*Reviewer #2 (Recommendations for the authors):*

In terms of presentation, I found the number and extent of data to be a bit overwhelming. If revising this paper, I hope the authors will consider trying to condense each figure to present a single message and summary panel and move data like how the number of contacts changes with time to the supporting information.

*Reviewer #3 (Recommendations for the authors):*

The authors have carried out a simulation study of the behavior of the TCR-pMHC complex for different peptides with and without load in the physiological range (10-20 pN). The load is calculated by applying harmonic restraints to the ends of the complex and extending their distances iteratively. The analysis of the complex under load seems to be novel.

Recommendations:

1) While the general conclusions regarding the load (e.g., higher number of contacts) seem to be supported by the simulations with different peptides, the conclusions regarding the different behavior of individual peptides (e.g., modified agonist vs. weak antagonist) are not fully supported as only one MD run was carried out for each peptide sequence and value of load. Multiple independent runs should be carried out for each simulation system, i.e., peptide and low/high load.

2) The definition of low and high load (Table 1) seems somewhat arbitrary as a low load of 13.2 pN and 14.9 pN is defined (from averaging over the 2nd half of the MD trajectory) for the WT peptide (full system and dFG, respectively) and these values are similar/higher than the high load of 13.5 pN of the P6A mutant.

[Editors' note: further revisions were suggested prior to acceptance, as described below.]

Thank you for resubmitting your work entitled "Asymmetric framework motion of TCRαβ controls load-dependent peptide discrimination" for further consideration by eLife. Your revised article has been evaluated by Qiang Cui (Senior Editor) and a Reviewing Editor.

The manuscript has been substantially improved and the reviewers appreciated the revision, but there are some remaining issues that need to be addressed, as outlined below:

A new comment concerns the standard deviations of the forces that have been added. The std are quite variable between the various structures (varying from ~2.5A to ~12A). I wonder whether this variability is interpretable in terms of the proposed catch-bond mechanism.

---

## [Author Response]

Essential revisions:1) From a technical point of view, discuss carefully the choice of using a restraining approach vs. applying a steady force, and clarify which better mimics the realistic situation.

Restraining the terminal C*_α_* atoms of the complex (blue spheres in Figure 2A) was done upon considering the realistic situation of immune surveillance. When a T-cell interacts with an antigen presenting cell (APC), other molecules such as CD2 and CD58 maintain the separation near the ∼120-°A span of the TCR*αβ*-pMHC complex (Reinherz,
*et al.,*
2023; see SI Text 3, lines 106–108 of the paper). The force applied to the complex fluctuates via thermal fluctuation and through cellular activities such as coupling to the actomyosin machinery within the T-cell and APC (Figure 3 of Reinherz,
*et al.,*
2023). Restraining terminal C*_α_* atoms in simulation mimics the membrane anchoring of these molecules with a relatively constant spacing and fluctuating force. In a constantforce simulation, in addition to the absence of fluctuation in force, the absence of positional restraint would not capture the situation where TCR and pMHC are anchored to the membranes of T-cell and APC, respectively.

Capturing effects of anchoring is important since bending and/or shearing loads can be applied to the TCR*αβ*pMHC interface, which would be much weaker if the whole complex is allowed to move transversely without any restraint (see also our reply R1-4 about this point).

A restrained distance constraint as suggested by Reviewer 1 (R1-6) would allow force to fluctuate, but it would not capture membrane anchoring. In addition, without any positional restraint, the simulation box will have to be made larger to accommodate rotational motion of the whole complex.

Even though applying positional restraints may better capture reality, it has practical issues: It is difficult to find the extension that yields a desired force level, for which we performed ‘laddering’ simulations that scan a range of extensions (page 18, ‘*Selecting extension*’). Since these initial simulations last only up to 100 ns at each extension, the average force after 500 ns in a 1-*µ*s production run at a selected extension can differ significantly. Issues with the positional restraints are further explained in R3-1.

We are currently testing different strategies of applying loads to find a more reliable and efficient simulation protocol, which would be a subject of a future publication. Despite methodological limitations, the main conclusions of our manuscript are well-supported by the data presented, as noted in the reviewers’ comments.

Changes made: The above are explained in:

– Pages 3–4, newly added Results section ‘*Applying loads to TCRαβ-pMHC complexes*:’ Explains our choice of applying positional restraints instead of constant forces.

– Page 18, lines 569–574: Expanded discussion about practical issues with applying positional restraints.

In addition, the principal component analysis can be done differently to ensure the most meaningful comparison between different cases.

Our main goal of performing principal component analysis (PCA) based on the V*α*-V*β* triads and the V-C beads-on-chain (BOC) models was to examine the relative motion among the 4 subdomains of TCR. To this end, we did apply PCA in different ways. To examine the V*α*-V*β* motion (Figure 4B), we aligned coordinate trajectories relative to C*_α_* atoms used for constructing the V*α*-V*β* triads (reference residues are listed in lines 650–651). To examine the V-C motion (Figure 5A), we aligned coordinate trajectories relative to the C-module (reference residues are listed in lines 672–673). Since there is very little C*α*-C*β* motion, the resulting PCA reveals the motion of the V-module relative to the C-module. Since different simulations involve the same A6 TCR, comparing the PCA across them reveals the effect of load and ligand on the conformational motion of TCR. This point is further addressed in our reply R2-3.

PCA is just one of several analyses we did. For example, the angles between triad arms (Figure 4C,D) and the V-C angle (Figure 5D,E) utilize the triads and BOC models, but they do not involve PCA. By using different approaches in combination, we find a consistent and physically plausible mechanism of the TCR catch bond and its ligand dependence.

Changes made: We added explanations about different ways of aligning coordinate trajectories:

– Page 6, lines 171–175: For V*α*-V*β* PCA.

– Page 8, lines 226–227 and caption of Figure 5A: For V-C PCA.

– Page 8, lines 203 and 238–239: Emphasized that angle analyses are independent of PCA.

2) In terms of results, discuss more explicitly the specific features that discriminate between catch-bond and slip-bond regimes. It is also valuable to explicitly suggest a set of experimentally testable predictions from the simulation study.

For TCR*αβ*-pMHC, catch bond does not activate through any defined conformational transition as observed in other adhesion proteins (e.g., integrin). Instead, our results show that it is achieved by load-dependent changes in the conformational motion of the protein that affect the stability of the interface. We measured the interfacial stability in multiple ways, including: Maintenance of the initial high-occupancy contacts (Figure 3B-E), increase in the buried area of the contact-forming residues (Figure 3G), stabilization of the CDR3 distance (Figure 4D), and related time dependent features (Figure 8). The newly added Figure 1 gives an overview of different measurements. Given the seconds-order bond lifetime in reality, in microseconds-long simulations, slip bonds would manifest as deterioration of the interfacial stability under higher forces but without any dissociation actually occurring during the simulation. The lower stability of the interface observed in the modified agonist or antagonist systems indicate they will exhibit weaker catch bond or slip bond behaviors. Single-molecule experiments of the A6 TCR system, which are currently not available, would be a good test of our simulation.

Other than the relatively straightforward predictions about modified agonists and antagonists, the allosteric catch bond mechanism we found is consistent with available experimental data. They include a reduced catch bond behavior of the C*β* FG-loop deletion mutant (Das
*et al.,*
2015), and a chimeric V*γδ*-C*αβ* receptor that attains catch bond capacity while wild-type TCR*γδ* works as a slip-bond receptor (Mallis,
*et al.,*
2021). Also, given the importance of the asymmetric motion, making mutations to weaken the V*β*-C*β* interface to decouple the V-module from the C-module would render the TCR to exhibit slip bond only. See also our reply R1-0 for additional explanation about making point mutations to TCR. However, making predictions about specific mutations would require additional extensive simulations, which is beyond the scope of the current work.

Changes made:

– Page 2: Figure 1 was added to provide an overview of different simulations and analyses (the rest of the figure numbers shifted by 1).

– Page 13, Concluding Discussion, lines 363–366 and page 14, lines 391–395: We added explanations about the main feature of the TCR*αβ* catch bond mechanism, as well as expected behaviors of catch- vs. slip bond systems in simulation.

– Page 14, Concluding Discussion, lines 384–385: We added a sentence explaining the agreement between simulation and experiment on the FG-loop deletion mutant.

– Page 14, Concluding Discussion, lines 387–390: Explanation about the V*γδ*-C*αβ* chimera was added and the reference (Mallis,
*et al.,*
2021) was added.

– Page 14, Concluding Discussion, lines 395–402: We added a discussion about using simulations for designing and testing point mutants.

Reviewer #1 (Recommendations for the authors):1. I suggest that the authors discuss why they chose to restrain the TCR/MHC separation, rather than devise an algorithm to apply a steady force. The issue with the restrained distance is that the forces reported for the different mutants are quite variable, and one might even say, not consistent.

As explained in our reply R0-1, we used positional restraints instead of applying constant forces to better reflect reality where the applied force fluctuates while TCR and pMHC are anchored to respective cell membranes separated by a relatively constant distance. Since fluctuation in the instantaneous force is rapid, the TCR*αβ*pMHC complex does not have time to respond at each moment. Yet, the fluctuating force drives the complex to respond differently depending on the average force, as demonstrated by our simulations. If a constant force were applied, the extension of the complex would fluctuate more widely, and the load actually experienced at the interface between TCR and pMHC may not be the same as the constant force applied to the ends, which would be similar to the case of fluctuating force under constant extension. A more detailed comparison between constant extension vs. constant force methods would be an interesting subject of a future study. For the present manuscript, we believe our main conclusions regarding the TCR catch bond mechanism are well-supported by constant-extension simulations.

Changes made:

Changes are the same as those listed in our response R0-1.

2. Uncertainty should be reported as the forces in some form, or the magnitude of their fluctuation.

We added standard deviations in forces measured in time intervals on page 4, Table 1 and page 27, Appendix 2–table 1.

3. (Table1) it might be interesting to check whether the integral of the force over the three extensions reported in the table correlates with the TCR/pMHC binding strength, if these data are available.

While this is an interesting idea, the work done by force in our simulation and the binding strength have nontrivial relation. For example, the dFG-pMHC system had a greater increase in force (14.9 pN to 29.0 pN) compared to the WT (13.2 pN to 18.2 pN) over similar change in extension (page 4, Table 1). This does not indicate that dFG-pMHC has a higher binding strength. On the contrary, the higher force in dFG-pMHC in the high-load case is due to the conformational change that shortened its longitudinal span, which would have a greater destabilizing effect in a longer simulation, as explained in Appendix 1 (page 27). To quantitatively address energetics, a much more extensive sampling simulation at incremental distances would be needed, with the added complexity of dealing with a catch bond.

Regardless of the practical difficulty, the energetics involved in antigen discrimination and TCR triggering is a fundamental issue, which we briefly mention in the revised manuscript.

Change made:

On pages 14–15, Concluding Discussion, lines 414–417, we mentioned about the energetics issue.

4. A few simulation observations either appear speculative or are not well illustrated, (e.g. on p5), "short distance between restraints … allows wider transverse motion that in turn generates a shear stress or a bending moment at the interface". Given the complexity of this large biomolecular complex and its dynamics, I suggest making a greater effort to distinguish between what is actually observed and what the implications might be.

We believe the revised manuscript incorporating reviewers’ comments have become clearer in providing reasoning behind key points. In particular, the newly added Figure 1 gives an overview of our different simulations and analyses, which helps with following individual Results sections. Regarding the example that the reviewer pointed out, it was based on the expectation that a loosely held molecule will transversely fluctuate more compared to a tightly held one. To check, we measured the root-mean square fluctuation (RMSF) of the center of mass of the C*_α_* atoms of the WT Tax peptide in the transverse direction. It was 16.3 °A for WT^low^ and 12.7 °A for WT^high^.

Mechanically, a wider transverse fluctuation while the ends are restrained is expected to increase the shear stress and/or bending moment, thereby promoting destabilization of the interface. A related measure of the relative motion between TCR and pMHC at the interface is the peptide angle relative to the variable domain which indeed varies more widely for WT^low^ than WT^high^ (Figure 8C).

Changes made:

– Page 2: Figure 1 was added to provide an overview of different simulations and analyses (the rest of the figure numbers shifted by 1).

– Page 6, lines 158–160: Transverse RMSF values of the peptide were added.

– Page 12, lines 349–351: A sentence was added to relate the wider fluctuation of the peptide angle in Figure 8C for WT^low^ to its greater transverse RMSF.

5. While there are various analyses of the simulation data, it would strengthen the paper greatly if the authors could provide specific experimentally testable hypotheses, eg., in the form of predicted responses to a mutant peptide, or mutations to the variable chains that could alter the fluctuations (e.g. disulfide crosslinking).

This is addressed in R0-3 and R1-0.

Reviewer #2 (Recommendations for the authors):In terms of presentation, I found the number and extent of data to be a bit overwhelming. If revising this paper, I hope the authors will consider trying to condense each figure to present a single message and summary panel and move data like how the number of contacts changes with time to the supporting information.

Given the extent of data and multiple analyses involved, we indeed put much efforts to simplify the data presentation and carefully selected which data to include as the main figures. Figure 8 that the reviewer suggested to move to the supporting information, is particularly important as it shows how forces and contact occupancy fluctuate over time, yet they reflect different loading conditions or ligand. Figure 8 also is relevant to the proposed model in Figure 9. Instead of further reducing figures, we added Figure 1 as the summary panel suggested by the reviewer. We hope it will help with grasping the flow of the manuscript.

Reviewer #3 (Recommendations for the authors):The authors have carried out a simulation study of the behavior of the TCR-pMHC complex for different peptides with and without load in the physiological range (10-20 pN). The load is calculated by applying harmonic restraints to the ends of the complex and extending their distances iteratively. The analysis of the complex under load seems to be novel.Recommendations:1) While the general conclusions regarding the load (e.g., higher number of contacts) seem to be supported by the simulations with different peptides, the conclusions regarding the different behavior of individual peptides (e.g., modified agonist vs. weak antagonist) are not fully supported as only one MD run was carried out for each peptide sequence and value of load. Multiple independent runs should be carried out for each simulation system, i.e., peptide and low/high load.

In addition to the data presented, we did carry out additional 2 replicate runs for the WT and 1 replicate run for each of the mutant systems. However, as was briefly explained in the Methods section of our original manuscript (page 18, lines 563–568) and in Appendix 2 for *in silico* mutants (page 27, lines 881–886), it was difficult to achieve monotonic increase of force with extension. This was because in some of the runs the Cterminal added strands made contact with the C-module of TCR, effectively shortening the span of the complex during the simulation. This led to the average force comparable to or higher than that for a larger extension. The runs included in Table 1 were those where low and high extensions resulted in low and high average forces (measured after 500 ns), respectively. Importantly, for other replicate simulations, we found that certain measures such as the CDR3 distance and average contact counts, are consistent with the average load rather than average extension, which agrees with the picture that the response at the interface is determined by the applied load.

Aside from replicate runs, simulations of *in silico* mutants (switching between the WT and mutants; Appendix 2 on pages 27–28) serve as additional tests, which yielded overall consistent results. Most of all, the asymmetric motion observed in all of the simulations and its impact on stability of the interface with pMHC provide a physically plausible explanation for the catch bond mechanism that does not rely critically on details of any single simulation.

Changes made:

– Page 13, lines 372–373 in Concluding Discussion: We added a remark about the generality of our findings obtained by analyzing multiple simulations collectively.

– Page 18, lines 560–574 in Methods: We explained about the issues we found about replicate simulations and the dependence of the responses on load rather than extension *per se.*

2) The definition of low and high load (Table 1) seems somewhat arbitrary as a low load of 13.2 pN and 14.9 pN is defined (from averaging over the 2nd half of the MD trajectory) for the WT peptide (full system and dFG, respectively) and these values are similar/higher than the high load of 13.5 pN of the P6A mutant.

This partly relates to the difficulty of precisely controlling forces at a given extension, as explained in R0-1 and R3-1. But even when the difference in forces is accounted for, the WT and mutant systems behave distinctly. In the case of the antagonist P6A that the reviewer pointed out, the average force of P6A^high^ (13.5 pN) is the lowest among the high-load simulations shown in Table 1. Both its extension and average force are comparable to those of WT^low^. If P6A^high^ were considered in isolation, it could be argued that the instability of the interface observed in simulation would be due to insufficient force, similar to WT^low^. However, when P6A^0^ (no load) and P6A^low^ are considered together, difference from the WT system emerges. This can be seen by the overall lower stability of the interface and weaker dependence on load (e.g., Figure 3A,G for WT vs. Figure 6A,B for P6A), differences in conformation; in Figure 7D, the average BOC for WT^low^ lies between those for WT^0^ and WT^high^, all of which differ substantially from the average BOCs for all P6A systems tested. Also, compared to WT^low^ that gradually loses contacts with pMHC, P6A^high^ has low contact occupancy early on in the simulation (Figure 8A). The lower stability of the interface is also evident for the *in silico* P6A mutant (^WT^P6A; see Appendix 2–figure 1A on page 28). As seen in Appendix 2-table 1 (page 27), loads on the *in silico* P6A are comparable to or higher than those used for WT in Table 1. Also note the average force on ^WT^P6A^low^ (24.7 pN) is higher than that on ^WT^P6A^high^ because the added strands made contacts with the TCR’s C-module in ^WT^P6A^low^ (page 27, lines 879–883 in Appendix 2; also explained in R3-1). Since we did not perform any replicate simulations for *in silico* mutants, we used these data directly.

Changes made:

– Page 4, lines 113–115, newly added Results section ‘*Applying loads to TCRαβ-pMHC complexes*:’ Added a remark that distinction can be seen between different systems despite variations in the applied load.

– Page 18, lines 569–574: Expanded discussion about practical issues with applying positional restraints.

[Editors’ note: what follows is the authors’ response to the second round of review.]

The manuscript has been substantially improved and the reviewers appreciated the revision, but there are some remaining issues that need to be addressed, as outlined below:A new comment concerns the standard deviations of the forces that have been added. The std are quite variable between the various structures (varying from ~2.5A to ~12A). I wonder whether this variability is interpretable in terms of the proposed catch-bond mechanism.

This is a very interesting observation. We examined the data more carefully, and found that there are both thermodynamic and system specific aspects to the large variation in the std of force. Regarding the thermodynamic aspect, the std tends to be larger for larger average forces. This can be seen more clearly by plotting all the values in Table 1 (Appendix 3-figure 1).

Except for Y8A^low^ and dFG^low^, all other data points lie on nearly a straight line. Thermodynamically, the force and position of the restraint (blue spheres in Figure 2A) form a pair of generalized force and the corresponding spatial variable in equilibrium at temperature 300 K, which is akin to the pressure *P* and volume *V* of an ideal gas.

If *V* is fixed, *P* fluctuates. Denoting the average and std of pressure as ⟨*P*⟩ and as ∆*P* respectively, Burgess showed that ∆*P/*⟨*P*⟩ is a constant (Eq. 5 of Burgess, Phys. Lett. A, 44:37; 1973). In the case of the TCR*αβ*-pMHC system, although individual atoms are not ideal gases, since their motion lead to the force fluctuation of the restraints, the situation is analogous to the case of an ideal gas where pressure arises from individual molecules hitting the confining wall as the restraint. Thus, the near-linear behavior in the figure above is a consequence of the system being many-bodied and at constant temperature. The linearity is also an indirect indicator that sampling of force was reasonable.

In addition to the thermodynamic aspect, system-specific aspects influence the std of force. In the above figure, Y8A^low^ is an antagonist that had the smallest number of contacts with pMHC except for Y8A^0^ without load (Figure 6A of our manuscript). dFG^low^ also had similarly small number of contacts with pMHC (Appendix 1figure 1A). The weakly held interface likely caused a wider conformational motion, leading to greater fluctuation in force relative to the average (dFG^low^ and Y8A^low^ in Figure 8A, symbols without outline).

The above suggest that the variation in the std of force *per se* does not provide a clear distinction between catch vs. slip bonds, although a comparatively larger std is indicative of potential instability. Feedback from the reviewers motivates us to carry out a future study focused on fundamental aspects of forces, constraints, and fluctuations in a smaller model system for more thorough analysis.

Changes made: